# A shape-shifting nuclease unravels structured RNA

Katarina Meze[1,2,3], Armend Axhemi[1,3], Dennis R. Thomas ®[1,3], Ahmet Doymaz[2,4] & Leemor Joshua-Tor ®[1,2,3] ✉

RNA turnover pathways ensure appropriate gene expression levels by eliminating unwanted transcripts. Dis3-like 2 (Dis3L2) is a 3′–5′ exoribonuclease that plays a critical role in human development. Dis3L2 independently degrades structured substrates, including coding and noncoding 3′ uridylated RNAs. While the basis for Dis3L2's substrate recognition has been well characterized, the mechanism of structured RNA degradation by this family of enzymes is unknown. We characterized the discrete steps of the degradation cycle by determining cryogenic electron microscopy structures representing snapshots along the RNA turnover pathway and measuring kinetic parameters for RNA processing. We discovered a dramatic conformational change that is triggered by double-stranded RNA (dsRNA), repositioning two cold shock domains by 70 Å. This movement exposes a trihelix linker region, which acts as a wedge to separate the two RNA strands. Furthermore, we show that the trihelix linker is critical for dsRNA, but not single-stranded RNA, degradation. These findings reveal the conformational plasticity of Dis3L2 and detail a mechanism of structured RNA degradation.

RNA quality control and turnover are vital for cellular function, yet little is known about how nucleases deal with the diverse universe of structured RNAs. Dis3-like 2 (Dis3L2) is an RNase II/R family 3′–5′ hydrolytic exoribonuclease that plays an important role in development and differentiation[1,2], cell proliferation[3–6], calcium homeostasis[7] and apoptosis[8,9] by effectively removing or processing 3′ uridylated RNAs[1,10–13]. Dis3L2 targets are oligouridylated by the terminal uridylyl transferases (or TUTs)[14–16]. The specificity toward uridylated RNAs is conferred through a network of base-specific hydrogen bonds along the protein's extensive RNA-binding surface, as demonstrated by the structure of *Mus musculus* Dis3L2 (MmDis3L2) in complex with a U₁₃ RNA[17].

Genetic loss of Dis3L2 causes Perlman syndrome, a congenital overgrowth disorder that is characterized by developmental delay, renal abnormalities, neonatal mortality and high rates of Wilms' tumors[1]. The first reported physiological substrates of Dis3L2 were the uridylated precursors of let-7 microRNAs[10,13], which play an important role in stem cell differentiation by silencing growth and proliferation

genes such as *HMGA2*, *MYC* and *Ras*[18–23]. Many other noncoding RNA targets have since been reported, including other microRNAs[24,25], transfer RNA fragments[16], small nuclear RNA[26], the intermediate of 5.8S ribosomal RNA processing 7S_B[27], the long noncoding RNA RMRP[28], and the 7SL component of the ribonucleoprotein signal recognition particle required for endoplasmic reticulum-targeted translation[7]. The latter is probably responsible for the Perlman syndrome phenotype, with aberrant uridylated 7SL leading to endoplasmic reticulum calcium leakage that perturbs embryonic stem cell differentiation, particularly in the renal lineage[7].

Unlike a number of structurally similar homologs, Dis3L2 can degrade structured RNAs independent of external helicase activity[1,10,12,29]. Little is known about how Dis3L2 or other capable RNase R/II family nucleases independently degrade structured RNA. We determined the structures of an RNase R/II family nuclease bound to a series of structured RNA substrates and analyzed the kinetic profiles of wild-type *Homo sapiens* Dis3L2 (HsDis3L2) and engineered mutants

[1]W.M. Keck Structural Biology Laboratory, Howard Hughes Medical Institute, New York, NY, USA. [2]School of Biological Sciences, Cold Spring Harbor Laboratory, New York, NY, USA. [3]Cold Spring Harbor Laboratory, New York, NY, USA. [4]Present address: Weill Cornell Medical College, New York, NY, USA. ✉e-mail: leemor@cshl.edu

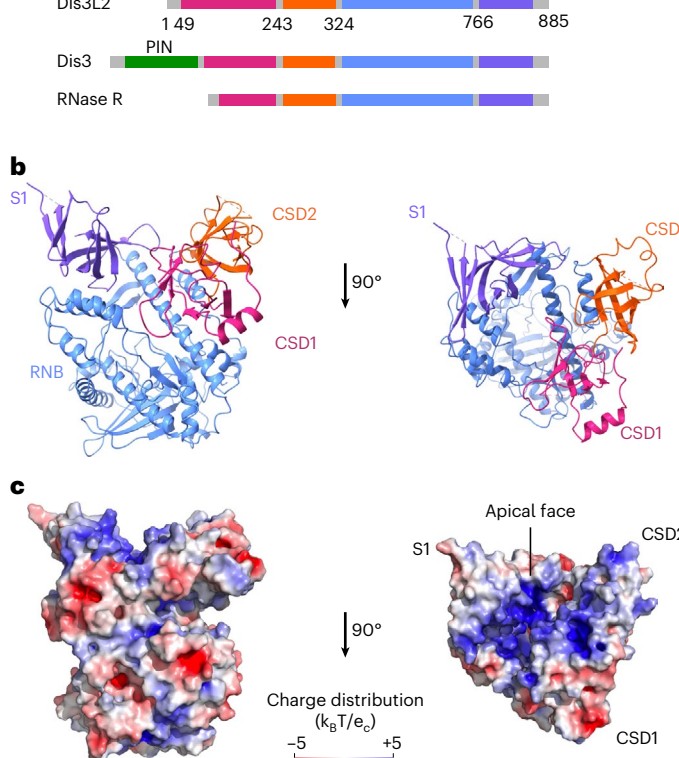

**Fig. 1 | RNA-free Dis3L2 is preorganized to bind RNA substrates. a**, Domain compositions of Dis3L2 and the homologous proteins Dis3 and RNase R (green, N-terminal PIN domain; pink, CSD1; orange, CSD2; blue, RNB; purple, S1 domain). **b**, Side (left) and top or apical (right) views of RNA-free Dis3L2^[D391N] with domain labels. **c**, Charge distribution of the Dis3L2 surface from a side view (left) and a view of the apical face (right), as calculated using PyMol APBS at an ionic strength of 150 mM (Methods) where $k_B$ is the Boltzmann constant, T is the temperature in degrees Kelvin and $e_c$ is the unit of charge.

to reveal how this nuclease achieves highly efficient degradation of structured RNA.

## Results

### Initial binding of Dis3L2 to structured substrates

To understand the presubstrate binding state, we used cryogenic electron microscopy (cryo-EM) to determine the structure of RNA-free HsDis3L2 to 3.4 Å resolution (construct Dis3L2^[D391N] residues 1–858: carboxy (C)-terminal truncation of residues 859–885; and an engineered catalytic mutation of Asp for Asn at residue 391 in Dis3L2) (see Methods, Fig. 1a,b and Extended Data Fig. 1a–c). RNA-free HsDis3L2 has a vase-like conformation in which three oligonucleotide/oligosaccharide-binding (OB) domains—two cold shock domains (CSDs) and an S1 domain—encircle a funnel-like tunnel that reaches into the Ribonuclease B (RNB) domain and leads to the active site (Fig. 1b and Extended Data Fig. 1d). The OB domains provide a large positively charged surface, which probably acts as a landing pad for the negatively charged RNA (Fig. 1c). The overall structure of RNA-free Dis3L2 is very similar to the structure of the mouse Dis3L2–ssRNA complex (MmDis3L2–U_{13}) (root mean square deviation (RMSD) = 1.2 Å, calculated over all Cα pairs)[17]. Thus, the apoenzyme is preorganized to bind single-stranded RNA (ssRNA).

To probe the initial binding of Dis3L2 to structured substrates, we designed a short hairpin RNA mimicking the base of the pre-let-7g stem, with a UUCG tetraloop for stability and a 3′ GC(U)_{14} (16-nucleotide)

overhang as the uridylated tail (hairpinA–GCU_{14}; Fig. 2a). The resulting 3.1 Å cryo-EM structure of the Dis3L2^[D391N]–hairpinA–GCU_{14} complex revealed that Dis3L2 maintains the same vase conformation as is observed in the RNA-free form (Fig. 2b and Extended Data Fig. 2). However, the double-helical stem of the RNA was not resolved, suggesting that double-stranded RNA (dsRNA) is not stably engaged by the nuclease upon initial substrate association. Nonetheless, the quality of the density allowed assignment of 15 of the 16 nucleotides of the single-stranded 3′ overhang. The RNA follows the same path as is seen in the MmDis3L2–U_{13} structure (RMSD = 0.8 Å over Cα atom pairs) and also forms numerous base-specific hydrogen bonds with the protein (Fig. 2c–f). As in the MmDis3L2–U_{13} structure, seven nucleotides at the 3′ end are buried in the RNB tunnel (Fig. 2f), which can only accommodate ssRNA.

### RNA double helix engagement by Dis3L2

To examine the structural changes occurring upon substrate processing, we shortened the 3′ overhang to 12 uridines and further modified the stem to increase stability (hairpinC–U_{12}; Fig. 3a). We obtained a 3.1 Å structure of wild-type HsDis3L2 with hairpinC–U_{12} in which the double-helical stem of the RNA hairpin is clearly visible and nestled between the two CSDs and the S1 domain (Fig. 3b,c and Extended Data Fig. 3). The overall conformation of Dis3L2 does not change compared with the hairpinA–GCU_{14} or RNA-free Dis3L2 (RMSD = 0.56 and 0.61 Å, calculated over all Cα pairs, respectively). The basal junction of the hairpin interacts with the S1 and CSD1 domains, while the apical loop interacts with CSD2 (Fig. 3c–e). At this point, when the 3′ overhang is 12 nucleotides long, the 5′ end at the double strand–single strand junction moves toward a loop in CSD1 (N76–H81) (Fig. 3d). While the true start of the duplex (C2–G21) lies slightly closer to the S1 domain, the U1 5′ overhang forms a wobble base pair with U23 near the N76–H81 loop in CSD1.

### Drastic conformational rearrangement before dsRNA unwinding

Next, we designed a substrate with an even shorter, seven-nucleotide, 3′ overhang (hairpinD–U_{7}), since this is the minimal ssRNA length needed to reach the active site from the opening to the tunnel (Fig. 4a). Cryo-EM analysis of wild-type HsDis3L2 in complex with hairpinD–U_{7} resulted in a 2.8 Å structure (Fig. 4b,c and Extended Data Fig. 4a–d). Strikingly, it is immediately evident that the conformation of this complex is markedly different from the vase conformation observed thus far (Fig. 4d–h). The two CSDs moved ~70 Å clear to the other side of the vase rim via a hinge in the linker region between CSD2 and the RNB (Fig. 4e and Supplementary Video 1). This resulted in a new conformation reminiscent of a prong when viewed from the side (Fig. 4c). This large rearrangement is accompanied by smaller conformational changes in the S1 and RNB domains. The S1 domain moves such that it angles toward the double helix where it forms new interactions with nucleotides C15 and G16 in the backbone of the double helix, while a loop in the RNB domain moves by 10 Å in response to the new positioning of the CSDs (Fig. 4g,h and Extended Data Fig. 4g). We confirmed that the prong conformation is not an inactive trapped state caused by the high (75%) GC content in hairpinD–U_{7} by testing the activity of Dis3L2 on this substrate as well as analyzing cryo-EM data of Dis3L2 in complex with hairpinE–U_{7}, which had a GC content of 50% (Extended Data Fig. 4e,f). Although we were unable to obtain a high-resolution reconstruction, the maps clearly show that Dis3L2 is in the prong conformation.

The two CSDs move as a block with their relative orientations unchanged (Extended Data Fig. 4h,i). An overlay of the RNA-free and hairpinD–U_{7} Dis3L2 structures shows that the 5′ strand of the dsRNA hairpin would clash with CSD1 if the CSDs would remain in their original position (Fig. 4f). Thus, it appears that upon shortening of the ssRNA overhang, the structured portion of the RNA substrate pokes the enzyme and triggers this large rearrangement. The consequence

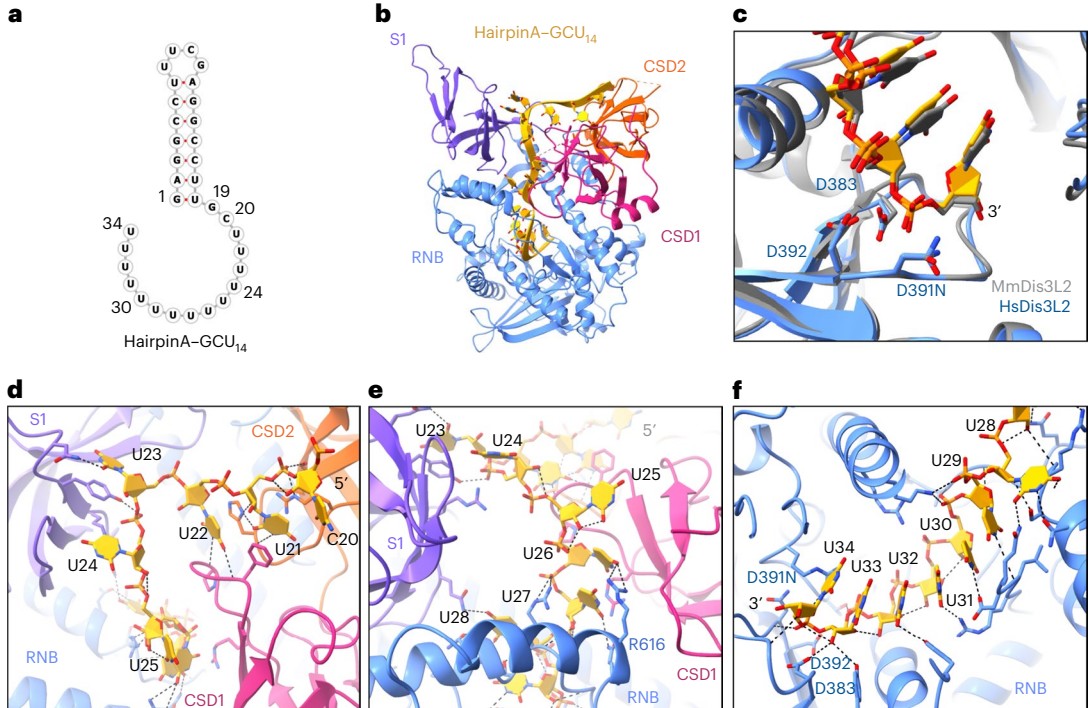

**Fig. 2 | Structured RNA is not engaged with Dis3L2 upon initial binding to the 3′ oligo-U tail. a**, RNA fold of hairpinA–GCU₁₄. **b**, Cryo-EM structure of Dis3L2^D391N in complex with hairpinA–GCU₁₄. **c**, Alignment showing the active site of MmDis3L2 in complex with U₁₃ (gray) (PDB: 4PMW) and HsDis3L2 in complex with hairpinA–GCU₁₄. **d**, The hydrogen bond network from C20–U25 traverses the apical face of the protein and involves both CSDs and the S1 domains. **e**, Following U25, the RNA enters into the narrow portion of the channel. **f**, The hydrogen bond network continues within the RNB all the way to the active site. For C20–U32, these include base-specific hydrogen bond interactions.

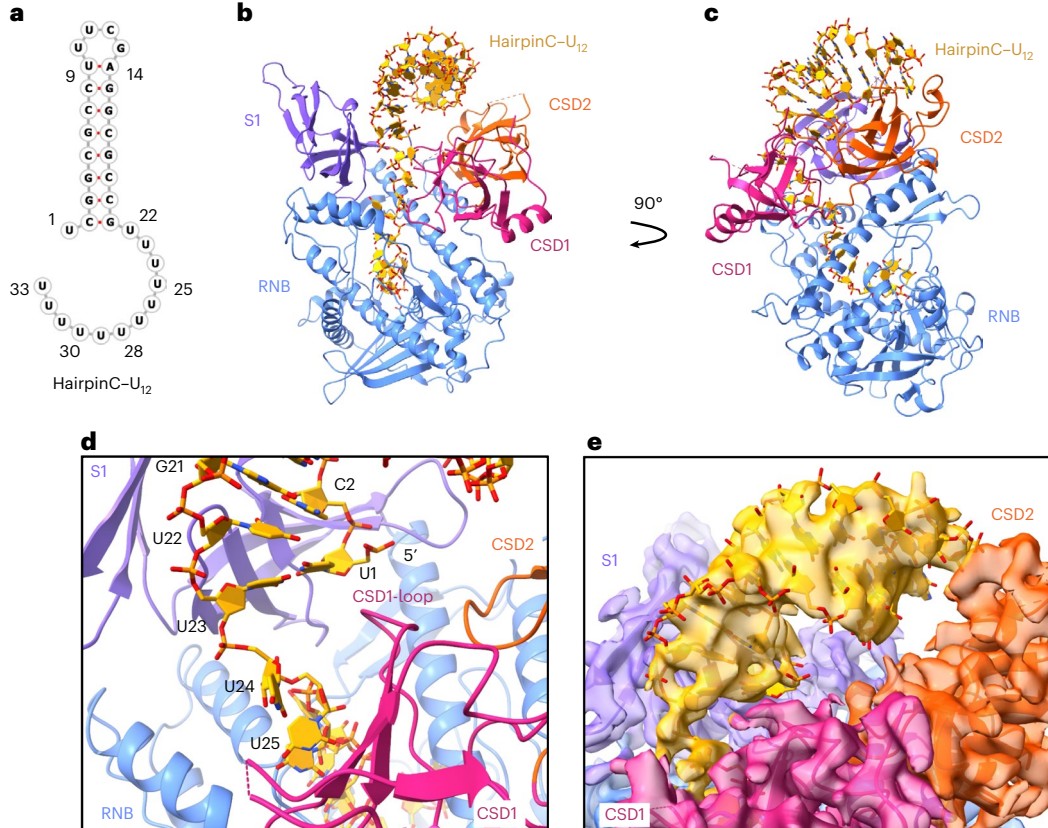

**Fig. 3 | The cryo-EM structure of HsDis3L2 in complex with hairpinC–U₁₂ shows engagement of the RNA duplex. a**, RNA fold of hairpinC–U₁₂. **b**, Wild-type Dis3L2 in complex with hairpinC–U₁₂. **c**, View of **b** at 90°. **d**, Basal junction of the double-helical stem and overhang. **e**, Overlay of the map and structure showing the position of the double-helical stem of hairpinC–U₁₂.

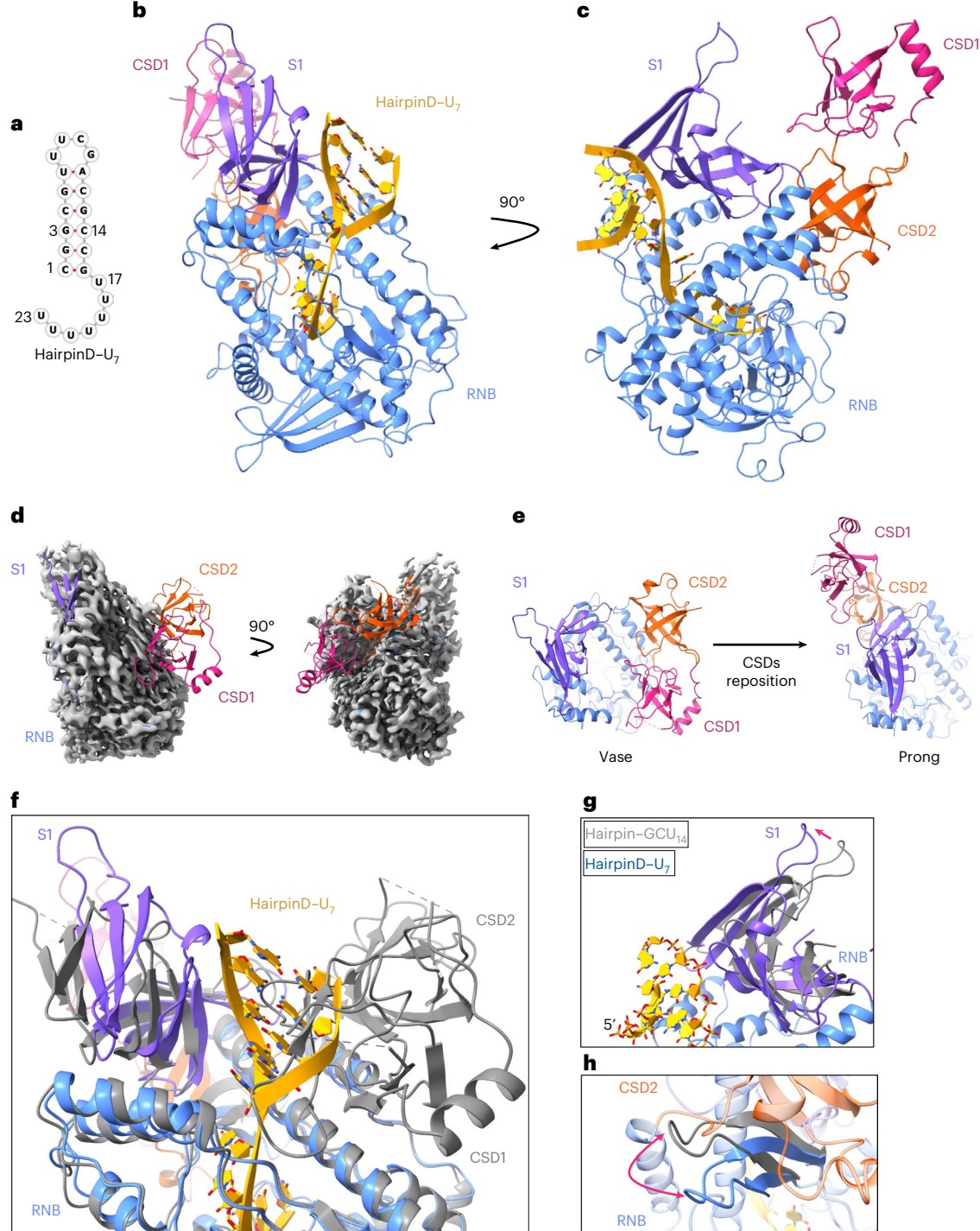

**Fig. 4 | Once the structured RNA gets closer to the enzyme, the CSDs reposition dramatically by 70 Å. a**, RNA fold of hairpinD–U₇ with a shorter, seven-nucleotide, 3' overhang. **b**, Structure of wild-type Dis3L2 in complex with hairpinD–U₇. **c**, A 90° view of **b**. **d**, Final 3D map of human Dis3L2 in complex with hairpinD–U₇, superimposed with the structure of RNA-free Dis3L2 fit in the map. **e**, View from the top, showing the change in position of the S1 and CSD domains in the vase and prong conformation. **f**, Alignment of RNA-free Dis3L2 (gray; vase) and Dis3L2–hairpinD–U₇ (colored domains and RNA; prong). The CSD domains are positioned behind S1 in the prong in this view. **g,h**, Alignment of hairpinA–GCU₁₄ and hairpinD–U₇ Dis3L2 structures showing the change in the positioning of the S1 domain (**g**) and a hairpin in the RNB (residues 555–572) (**h**).

of this movement is that it allows the structured portion of the RNA to come into contact with the RNB while also shortening the length of the narrow tunnel to the active site by two nucleotides (Fig. 5 and Extended Data Fig. 5a). Moreover, the RNA double-helical stem is now positioned on top of the junction between a bundle of three RNB helices and a linker connecting them to the rest of the RNB (Fig. 5b,c and Extended

Data Fig. 5). This junction would then act as a wedge to separate the two strands of RNA, allowing the 3' strand to enter into the narrow tunnel while the 5' strand peels away. Six out of seven residues in the single-stranded overhang are in the same position as in the hairpinA–GCU₁₄ and hairpinC–U₁₂ structures, with five fully buried in the tunnel of the RNB (Fig. 5c,d). There is no change in the final approach to the

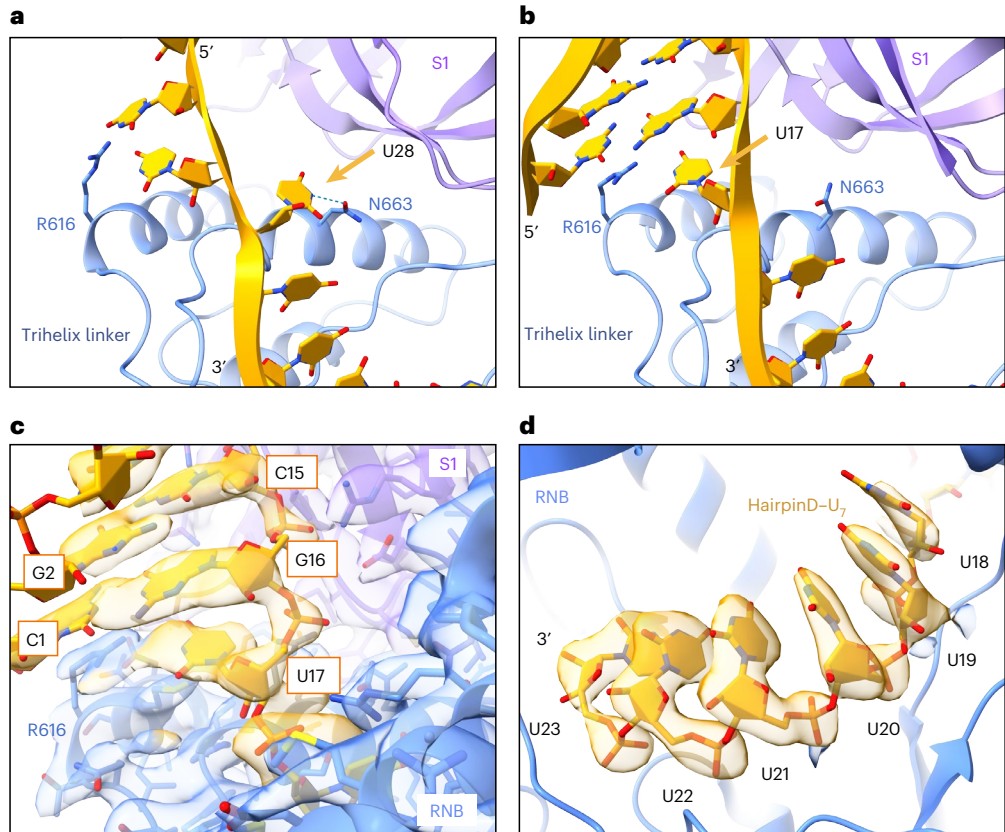

**Fig. 5 | The base of the double-helical stem is positioned above an RNB trihelix linker. a,b,** Comparison of RNA conformations near the trihelix linker in the hairpinA–GCU$_{14}$ (**a**) and hairpinD–U$_7$ structures (**b**). The seventh base from the 3′ end (U28 in hairpinA–GCU$_{14}$ and U17 in hairpin–U$_7$; yellow arrow) swings from interacting with the side chain of N663 (hairpinA–GCU$_{14}$) to pointing towards

R616 (hairpinD–U$_7$), which stacks under C1. The trihelix linker forms the final barrier to the dsRNA before the tunnel to the active site. **c,** Overlay of the cryo-EM map of the HsDis3L2–hairpinD–U$_7$ structure at the hairpin basal junction. **d,** Nucleotides U19–U23 are buried in the tunnel of the RNB.

active site. However, the seventh base from the 3′ end, which is also the first single-stranded base in hairpinD–U$_7$ (nucleotide U17), is no longer pointing towards the S1 domain and N663 as it is in the vase conformation (nucleotide U28 in hairpinA–GCU$_{14}$) (Fig. 5a). Instead, it flips to stack underneath G16 of the double-stranded stem and forms a pseudo base pair with R616, which emanates from the start of three α helices in the RNB (Fig. 5b,c).

Upon close examination of our cryo-EM data, we noticed that heterogeneous refinement yielded not only the high-resolution structure of the prong, but also a smaller three-dimensional (3D) class representing the vase (Extended Data Fig. 6a). Using a standardized analysis, we examined cryo-EM data of Dis3L2 with a series of substrates with identical stem loops but of varying single-stranded overhang lengths and looked at their particle distributions between 3D classes after heterogeneous refinement (Methods and Extended Data Fig. 6b–d). This analysis revealed the point at which the drastic change between the vase and prong conformation occurs. The vase conformation is the only one observed in the RNA-free form and with long single-stranded 3′ overhangs (Fig. 6 and Extended Data Fig. 6c). The prong conformation is first observed when the overhang is eight nucleotides long and is the only conformation observed when the overhang length is shortened to five nucleotides (Fig. 6 and Extended Data Fig. 6c). This illustrates the shape-shifting nature of the enzyme to enable the degradation of structured RNA substrates, and suggests that this dramatic conformational change is triggered by the RNA when the overhang is roughly eight nucleotides long. A similar distribution is seen for independent datasets (Extended Data Fig. 6c).

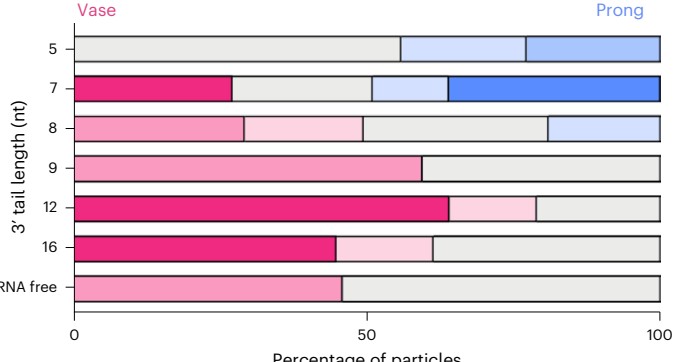

**Fig. 6 | Dis3L2 undergoes a conformational change at eight-nucleotide overhang lengths.** Distribution of particles after heterogeneous refinement for select datasets. The *x* axis shows the percentage of particles in the vase (pink) or prong (blue) conformation. The *y* axis shows individual datasets of RNA-free or hairpin RNA-bound Dis3L2, with numbers denoting the length of the 3′ overhang in nucleotides (nt). The deeper color indicates higher-quality 3D reconstructions, whereas gray indicates particles that did not contribute to a meaningful reconstruction.

**Dis3L2 degrades structured substrates with high processivity**

To quantitatively understand how the structural features described above impact the function of Dis3L2, we carried out presteady-state

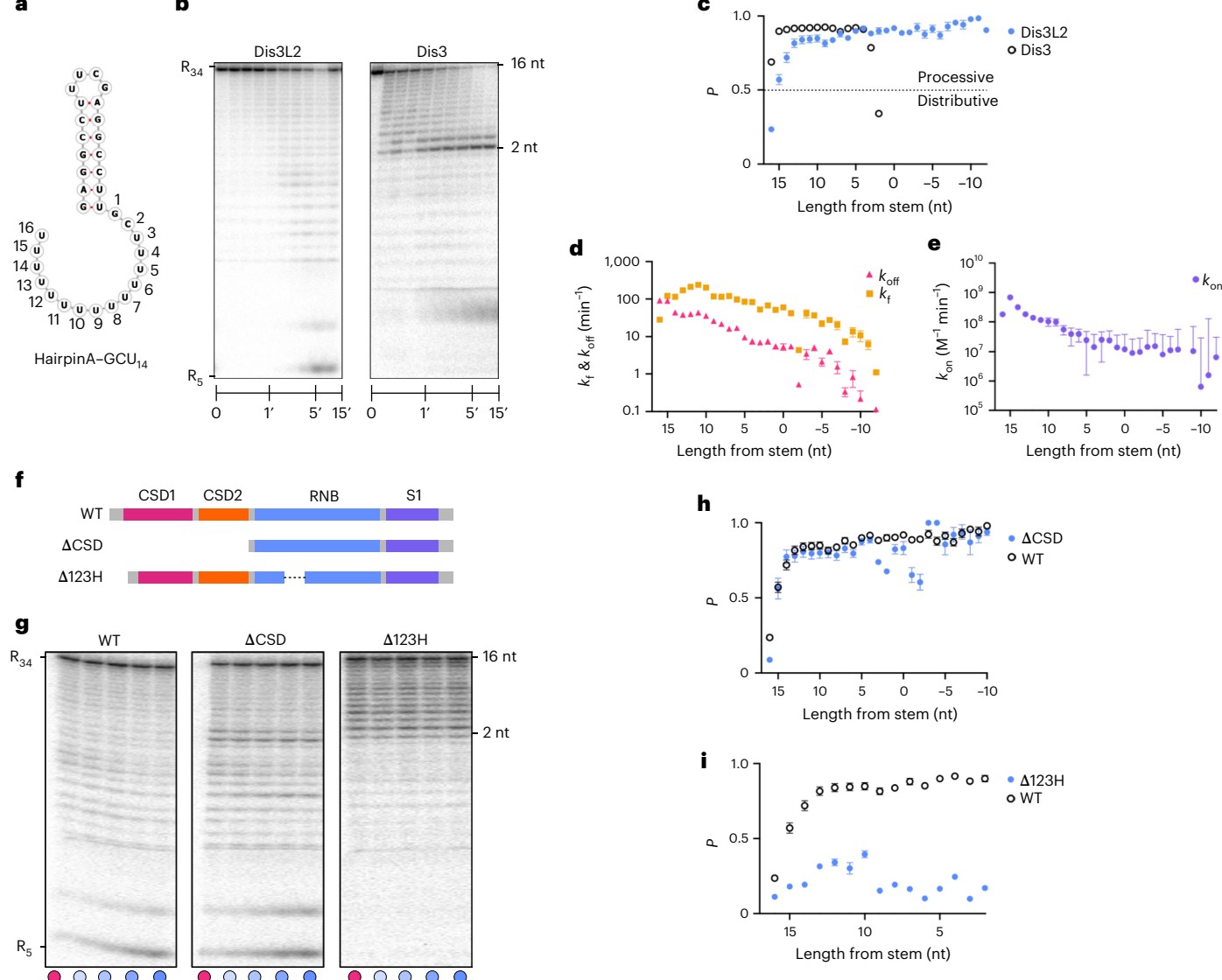

**Fig. 7 | Kinetic profile of structured RNA degradation by wild-type HsDis3L2 and mutants at single-nucleotide resolution. a**, Schematic of hairpinA–GCU$_{14}$. For simplicity, the kinetic data are numbered from 16 (3′ end) to 0 (single strand–double strand junction) to denote the nucleotide position. **b**, Two representative gels from presteady-state nuclease titration assays with 1 nM 5′ P$^{32}$-radiolabeled hairpinA–GCU$_{14}$ and 25 nM HsDis3L2 and HsDis3, respectively. The overall lengths of the species and the single-stranded overhang lengths are indicated on the left and right of the panels, respectively. Each experiment was repeated independently at multiple concentrations of each enzyme with $n$ = 2. **c**, Processivity ($P$) of wild-type Dis3L2 versus Dis3. **d**, Dissociation rate constants ($k_{off}$) and forward rate constants ($k_f$) of Dis3L2. **e**, Association rate constants ($k_{on}$) of wild-type human Dis3L2. $k_{on}$ data point $x$ = −8 was removed due to a large uncertainty value. The $x$ axis shows the number of nucleotides from the start of the double-stranded stem. **f**, Domain composition of wild-type (WT) human

Dis3L2 and the ΔCSD and Δ123H deletion mutants. **g**, Representative gels from pulse–chase reactions of wild-type human Dis3L2, ΔCSD and Δ123H at a 50 nM concentration with 1 nM radiolabeled hairpinA–GCU$_{14}$. Cold chase was added to the reaction at the 3-min timepoint to a final concentration of 5,000 nM. Measurements were taken prechase at 3 min (pink dot) and postchase at 4, 5, 7.5 and 10 min (blue gradient dots; light to dark, respectively). Each experiment was repeated independently at two concentrations of each enzyme with $n$ = 3. **h**, Processivity ($P$) of wild-type Dis3L2 versus ΔCSD. **i**, Processivity ($P$) of wild-type Dis3L2 versus Δ123H. The error bars in plots **d** and **e** represent s.e.m. from the global fit of data from the enzyme titrations (nine Dis3L2 concentrations with $n$ = 5) and pulse–chase experiments (two Dis3L2 concentrations with $n$ = 4). The error bars for the processivity plots in **c**, **h** and **i** show propagated errors calculated from the s.e.m. of $k_f$ and $k_{off}$ derived from the same global fit of the data.

kinetic assays and pulse–chase experiments to measure processivity ($P$) and elementary rate constants for RNA binding (association ($k_{on}$) and dissociation ($k_{off}$)) and degradation (forward step ($k_f$)) for wild-type and mutant forms of Dis3L2 at single-nucleotide resolution (Extended Data Fig. 7a–g). Our global kinetic analysis also provided a measure for the contribution of nonproductive binding (that is, substrate association that is not conducive to RNA degradation) (Extended Data Fig. 7a,e).

Using a 5′ $^{32}$P-radiolabeled 34-nucleotide hairpin RNA with a seven-base pair stem and a 16-nucleotide overhang (hairpinA–GCU$_{14}$) as a substrate (Fig. 7a), we found that wild-type Dis3L2 is distributive for the first step ($P$ = 0.23 ± 0.003), requiring approximately three binding events before cleaving the first nucleotide (Fig. 7b,c). This may serve as an important checkpoint before initiation of processive degradation in the second phase, where multiple nucleotides are cleaved before a dissociation event (Fig. 7c). This pattern was also observed in

substrates where the terminal base pairs were switched from GU and AU to GC and CG, respectively (hairpinB–GCU$_{14}$), or when the overhang was composed purely of Us (hairpinA–U$_{16}$) (Extended Data Fig. 7h–j). In the case of hairpinB–GCU$_{14}$, an interesting decrease in processivity was seen at an overhang length of five nucleotides, which may reflect some stalling before dsRNA unwinding (five to four nucleotides).

During the course of the reaction, as the RNA is progressively shortened from the 3′ end, the $k_{off}$ decreases (Fig. 7d). We observe a steeper decline in the $k_{off}$ starting around 11 nucleotides, which could reflect additional stabilizing interactions that become available when the phosphate backbone of the double-helical portion of the substrate is brought into contact with the enzyme (as was observed in the hairpinC–U$_{12}$ and hairpinD–U$_7$ structures). The $k_{on}$ also decreases with RNA length, probably due to the loss of substrate interaction points (fewer 3′ U binding sites) necessary for stable association (Fig. 7e). Interestingly, the $k_f$ varies with the stage of substrate degradation (Fig. 7d). After an initial slow step, $k_f$ increases as the enzyme degrades through the single-stranded overhang. When the overhang is shortened to 11 nucleotides, $k_f$ peaks and then begins to decrease as the enzyme encounters the dsRNA. This suggests that catalysis and/or translocation slow down as the dsRNA is engaged and later unwound, as $k_f$ reflects the slower of these two processes. Overall, $k_{off}$ has the dominant effect on processivity across all intermediate species (Fig. 7c).

When the substrate RNA initially associates with Dis3L2, it can do so productively (allowing degradation to commence from the 3′ end) or nonproductively. Examples of the latter include: the terminal nucleotide not binding fully into the active site; misorientation of the RNA; or a nonproductive/inactive conformation of Dis3L2. We observe roughly tenfold tighter nonproductive compared with productive binding for the very first step (Extended Data Fig. 7e). To confirm that the nonproductively bound species are not due to noncleavable RNA, we performed long-time-course experiments and showed that all species do get degraded to the five- to four-nucleotide end product (Extended Data Fig. 7b–c). This suggests that along with mediating association with substrates targeted for degradation, Dis3L2's RNA-binding regions may play a role in non-nucleolytic RNA-binding functions.

To assess whether these observations could be applicable to other substrates, we carried out additional processivity analyses of Dis3L2 degradation on two hairpins (hairpinF–U$_{16}$ and hairpinG–U$_{16}$) that mimic the 3′ end of the 7SL RNA (a natural substrate of Dis3L2) and one hairpin with a longer stem (hairpinI–GCU$_{14}$) (Extended Data Fig. 8). In all three cases, we observe a distributive first step followed by a processive phase, in line with our global observation of the processivity on hairpinA–GCU$_{14}$. Although some minor differences in the kinetic profile of these substrates are observed, the overall pattern remains. The magnitude of the first distributive step and the rate by which Dis3L2 reaches high processivity are different, however. The 7SL mimic hairpinG–U$_{16}$ has the lowest first-step processivity ($P = 0.16 ± 0.038$) and takes the longest to reach the very high processivity seen with hairpinA–GCU$_{14}$ in the second phase, possibly due to the bulkiness of this substrate. The 7SL mimic hairpinF–U$_{16}$ ($P_{16} = 0.21 ± 0.026$) and hairpinI–GCU$_{14}$ ($P_{16} = 0.28 ± 0.080$), which both harbor a single hairpin, have a kinetic profile more similar to hairpinA–GCU$_{14}$. Nevertheless, the overall features for all these substrates are similar.

We compared the kinetic profile of HsDis3L2 with that of HsDis3—an exosome-associated nuclease of this family that, in contrast with Dis3L2, cannot independently degrade structured substrates (Fig. 7b,c and Extended Data Fig. 9)[12]. HsDis3 appears to bind the oligo-U-tailed substrate much tighter and enters directly into processive degradation without an initial distributive step (Extended Data Fig. 9b,c). It maintains high processivity up until the single-stranded overhang reaches three to two nucleotides in length, at which point there is a drastic decrease in processivity as a result of a large increase

in the dissociation rate ($k_{off}$) (Fig. 7b and Extended Data Fig. 9b,e). This shows that, unlike HsDis3L2, HsDis3 is not able to maintain sufficient association with the substrate once it encounters the structured portion of the substrate.

## CSDs play multiple roles in RNA processing

Since the CSDs appeared to be the initial recognition sites for the RNA, but then triggered to move to the other side of the protein upon RNA processing, we tested whether they contribute predominantly to initial substrate association or ssRNA degradation. Removal of the CSDs (ΔCSD; deletion of residues 1–365) led to lower processivity for the very first step, largely due to a lower forward rate constant, indicating that the CSDs play a role in augmenting the rate of catalysis for the first nucleotide cleavage (Fig. 7f–h and Extended Data Fig. 10a–c). Furthermore, our analysis has shown that the CSDs provide roughly half of the nonproductive binding affinity (nonproductive $K_{1/2}$ = 4.2 ± 0.41 nM (wild type) versus 8.5 ± 0.73 nM (ΔCSD)) and removing them improves the productive binding fivefold (productive $K_{1/2}$ = 508.3 ± 1.27 nM (wild type) versus 97.7 ± 0.35 nM (ΔCSD)). During the following ssRNA degradation steps, ΔCSD has a similar processivity to wild-type Dis3L2 (Fig. 7h). However, there is a substantial decrease in the processivity as ΔCSD approaches the structured part of the RNA, showing a marked reduction at the three- and two-nucleotide single-stranded overhang position, as well as at the −1 and −2 nucleotide positions, which now fall within the RNA stem. This is a result of a notable increase in the dissociation rate ($k_{off}$) (Fig. 7h and Extended Data Fig. 10b). Thus, the CSDs contribute to both initiation of RNA degradation and maintenance of substrate association during the initial unwinding steps.

## The RNB trihelix and linker are necessary for resolving dsRNA

The Dis3L2–hairpinD–U$_7$ complex structure shows that the trihelix linker provides the final barrier before the narrow tunnel to the active site, suggesting a role in dsRNA unwinding. Deletion of the trihelix and linker (residues P612–M669: Δ123H) has a striking effect on substrate degradation and a buildup of intermediate species is observed at lengths close to the start of the double strand of hairpinA–GCU$_{14}$ (Fig. 7f,g,i). Δ123H never reaches the processive phase, although there is a slight increase in the processivity during initial degradation of the ssRNA overhang. When the substrate shortens to ten nucleotides in the overhang, the dissociation rate ($k_{off}$) increases significantly and the forward rate ($k_f$) plateaus, leading to a dramatic drop in processivity and a buildup of species with three- and two-nucleotide overhangs (Extended Data Fig. 10d–f). However, no such buildup was observed in the case of a single-stranded U$_{34}$ substrate, or with Dis3L2 mutants in which only one of the three helices was deleted (Extended Data Fig. 10g–k). This shows that the trihelix linker module as a whole is crucial for dsRNA, but not ssRNA, degradation.

## Discussion

Dis3L2 has emerged as a key nuclease responsible for the specific targeting and degradation of cytoplasmic uridylated RNAs, many of which are highly structured. Given that many exoribonucleases employ the help of helicases to degrade structured substrates, Dis3L2's ability to independently degrade dsRNAs with high processivity is mechanistically interesting. However, little was known about how Dis3L2 or other capable RNase R/II family nucleases achieve independent degradation of dsRNA. Here, we discovered a large conformational change that is triggered by the dsRNA and exposes a trihelix linker module that is crucial for the degradation of dsRNA but not ssRNA. We observed engagement of the dsRNA by Dis3L2's OB domains and uncovered an important contribution of the CSDs to initial nucleotide cleavage and duplex unwinding. We also identified the contribution of nonproductive binding. Below we discuss the implications of our findings for Dis3L2's mechanism of action and the function of other RNase R/II nucleases.

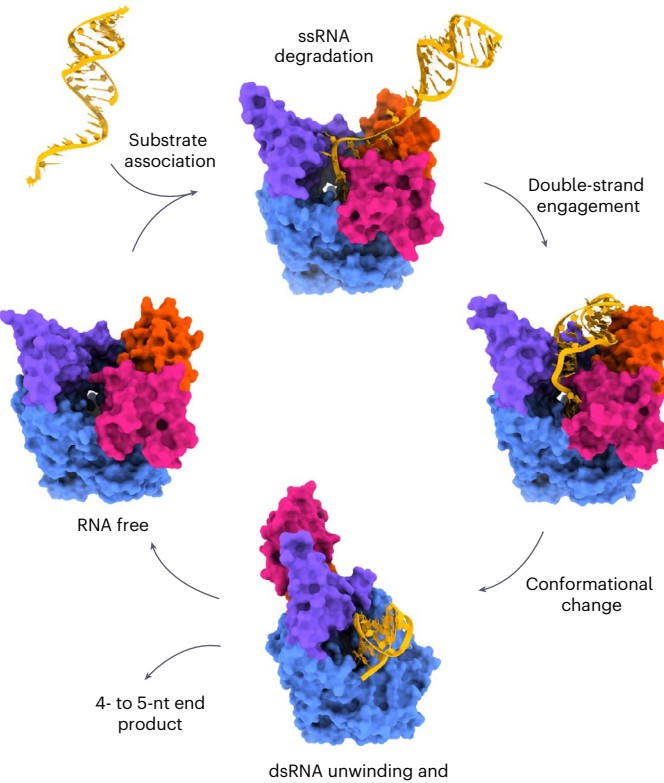

**Fig. 8 | Model of structured RNA processing by Dis3L2.** RNA-free Dis3L2 is preorganized in a vase conformation to bind RNA substrates (yellow), with a seven-nucleotide-deep tunnel leading to the nuclease active site. When the RNA overhang is shortened to ~12 nucleotides, additional contacts are made to the dsRNA. Further shortening of the overhang triggers a large rearrangement of the two CSD domains (pink and orange) to the prong conformation and allows the base of the dsRNA to access a module in the RNB domain (blue) that acts as a wedge to separate the two RNA strands and allows entry of one of the strands into the narrow tunnel leading to the active site. In this way, the enzyme ensures continued RNA degradation during RNA duplex unwinding.

### Role of the CSDs in the degradation of structured RNA substrates

Analysis of a Dis3L2 mutant in which the CSDs had been removed, ΔCSD, showed an impact on both the first catalytic step and the initial unwinding phase (Extended Data Fig. 10a,b). Moreover, before duplex unwinding, the dissociation constant increased (Extended Data Fig. 10b). Our structural data show that the CSDs switch to the prong conformation at overhang lengths of roughly eight nucleotides (Fig. 6), which makes the fact that we see an impact on binding during the unwinding phase in the ΔCSD variant somewhat confusing. There are two models that might explain these observations. The CSDs could be contributing to binding indirectly, by stabilizing the repositioning of the S1 domain to directly interact with the RNA double helix in the prong conformation. Alternatively, the CSDs may partially swing back to bind the dsRNA directly in a modified vase conformation (the CSDs would clash with the 5' strand in a full vase conformation). Since cryo-EM analysis showed that a hairpinD–U₅ substrate led to the prong conformation alone, the former model seems more likely. However, active turnover conditions may allow for more dynamic back-and-forth movement of the CSDs during degradation.

### Nonproductive binding and non-nuclease roles of Dis3L2

An example of nonproductive binding was observed in the crystal structure of MmDis3L2 (ref. [17]). While the RNA substrate provided was only 13 nucleotides in length, electron density for 14 nucleotides was

observed. This was due to two different positions of the RNA: with the 3' end either right in the active site (in a productive configuration) or removed away by one nucleotide, leaving the active site open. The latter position represents a nonproductive state. Nonproductive binding has been measured in other nucleases such as RRP6 (ref. [30]).

The large contribution from nonproductive binding might indicate that Dis3L2 functions in other roles that do not require nuclease activity. A recent study of hepatocellular carcinoma appears to have identified one such case[3]. Dis3L2 was found to be highly expressed in hepatocellular carcinoma tissues and promoted alternative splicing of the *Rac1* gene through a nuclease-independent mechanism. Dis3L2 was shown to bind the Rac1 pre-messenger RNA via the S1 domain and recruit heterogenous nuclear ribonucleoprotein (hnRNP)−U through its CSDs. This enabled the production of Rac1b, an isoform that promotes transformation and tumorigenesis[3].

Evolutionary analysis of Dis3L2 has revealed that the protein has lost nuclease activity at least four times during fungal evolution, while the CSDs have remained conserved, thereby suggesting a role for the protein outside of RNA degradation[31]. The Dis3L2 homolog in *Saccharomyces cerevisiae*, Ssd1, is an example of this, losing both canonical RNase II/R catalytic residues and acquiring a loop insertion that blocks the tunnel to the active site. Ssd1 has been reported to act as a translational repressor of certain messenger RNAs involved in cell growth and cytokinesis, and deletion of Ssd1 was found to have pleiotropic effects on stress tolerance[31,32].

### A comprehensive model for structured RNA degradation

Combining our cryo-EM and kinetic data, we propose the following model: RNA degradation by Dis3L2 proceeds via a minimum of six sequential steps: (1) substrate association and quality control; (2) initial nucleotide cleavage; (3) 3' single strand degradation; (4) double strand engagement; (5) dramatic domain realignment; and (6) concurrent double strand unwinding and degradation (Fig. 8). During the first four stages, Dis3L2 is in the vase conformation, with the S1 domain and CSDs positioned to form a large, positively charged surface for the oligo-U tail of the RNA. While the S1 and RNB domains provide crucial binding interactions, the CSDs enable effective initiation of degradation by contributing to the first catalytic step. This initial step is slow and acts as a substrate checkpoint. Once cleared, the enzyme enters the highly processive phase. When the overhang is shortened to 11 or 12 nucleotides, the RNA duplex engages with the enzyme, stabilized by contacts with the S1 domain and CSDs. At this point, the forward rate constant begins to decrease as the base of the dsRNA hairpin gets closer to the tunnel in the RNB domain. When the single-stranded 3' overhang reaches nine or eight nucleotides, the 5' strand of the RNA double helix runs into CSD1 and triggers a large movement of the two CSDs to the other side of the enzyme (see also Supplementary Video 1). In the resulting prong conformation, the S1 domain angles toward the tunnel and engages the backbone of the RNA double helix, which now sits over the RNB trihelix linker. The trihelix linker module acts as a wedge between the two RNA strands to separate them and enable the 3' strand to enter into the narrow part of the now shortened tunnel. Strand unwinding probably initiates when the overhang reaches roughly five nucleotides. Alignment of the structure of Dis3L2 in complex with hairpinD−U₇ with known structures of RNase R/II family nucleases suggests that most would have to undergo a similar conformational change to allow the double-stranded portion of the RNA access to the trihelix linker wedge. Biochemical studies of *Escherichia coli* RNase R have also demonstrated the importance of the trihelix in dsRNA degradation[33], suggesting that the mechanism proposed here could be conserved in other members of the RNase R/II family of nucleases. Collectively, this work unveils a molecular mechanism for efficient, regulatory degradation of structured RNAs by a vital nuclease.

## Online content

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

## Methods

### Protein preparation

Full-length human Dis3L2, mutants with domain deletions, point mutations, and HsDis3 were cloned as amino (N)-terminal Strep-Sumo-TEV fusion proteins in a pFL vector of the MultiBac baculovirus expression system[34]. Benchling (https://www.benchling.com) was used for sequence analysis and primer design. Expression and purification followed a similar protocol to that detailed by Faehnle et al.[17]. All constructs were expressed in SF9 cells grown in HyClone CCM3 Media (Thermo Fisher Scientific) at 27 °C for 60 h. Cells were then pelleted and resuspended in wash buffer (50 mM Tris (pH 8), 100 mM NaCl and 5 mM dithiothreitol (DTT)) and a protease inhibitor cocktail was added before snap freezing with liquid $N_2$ for storage at −80 °C. After thawing, cells were lysed by increasing NaCl to 500 mM, followed by one round of sonication. 0.1% poly-ethylene imine was added to the lysate and cell debris were cleared by 45 min of ultracentrifugation at 35,000 r.p.m. and 4 °C. The solution was then incubated with Strep-Tactin Superflow resin (IBA BioTAGnology) for 30 min while on a rolling shaker. The slurry was applied to a gravity column and washed with 20 column volumes of wash buffer before eluting the protein with 2 mM desthiobiotin in wash buffer. The Strep-Sumo-TEV tag was cleaved using TEV protease overnight at 4 °C. Cleavage efficiency and sample purity were assessed by sodium dodecyl sulfate polyacrylamide gel electrophoresis. The protein was then diluted to a final salt concentration of 50 mM in 25 mM HEPES (pH 7.5) and 5 mM DTT and applied to a HiTrap Heparin HP affinity purification column (GE Life Sciences) equilibrated in 50 mM NaCl, 25 mM HEPES (pH 7.5) and 5 mM DTT. The bound protein was eluted by applying a linear increasing salt gradient (0.05–1 M NaCl). Pooled fractions of protein were then concentrated and loaded onto a 10/300 Superdex 200 Increase gel filtration column (GE Life Sciences) equilibrated in 20 mM HEPES (pH 7.5), 150 mM NaCl and 5 mM DTT. Protein purity was assessed by the quality of the chromatogram and by running sodium dodecyl sulfate polyacrylamide gel electrophoresis gels. The concentrated sample was frozen in liquid $N_2$ and stored at −80 °C.

### Cryo-EM sample and grid preparation

RNA oligos were purchased from Dharmacon, then RNA secondary structure predictions were done using the Vienna RNAfold web server (http://rna.tbi.univie.ac.at) and diagrams were made using Forna (Figs. 2a, 3a, 4a and 7a and Extended Data Figs. 4a and 8a)[35–37]. RNA hairpins were annealed by diluting into 20 mM HEPES (pH 7.5), 150 mM NaCl and 5 mM DTT and heated to 95 °C for 3 min before stepwise cooling (95, 50, 30 and 4 °C). Complexes of human Dis3L2 and various RNAs were prepared by mixing equimolar ratios of Dis3L2 and RNA, incubating for 15 min and loading onto a 10/300 Superdex 200 Increase gel filtration column (GE Life Sciences) equilibrated in the same buffer (for specific controls indicated in Extended Data Fig. 6c, 100 μM ethylenediaminetetraacetic acid (EDTA) was added to the buffer). Complex formation was evaluated by monitoring a peak shift and the ratio of absorbance at 260 and 280 nm. Fractions of the complex were then pooled and concentrated to roughly 0.5 mg ml⁻¹ for Quantifoil carbon-coated Cu grids or 0.3 mg ml⁻¹ for Au foil grids (Quantifoil). Next, 4 µl of sample was applied to glow-discharged grids and a Vitrobot plunger (Thermo Fisher Scientific) was used to freeze the grids in liquid ethane (95% humidity; 20 °C; blot force 4; blot time 2.5 s).

### Cryo-EM data acquisition and image processing

Data were collected on a 300 kV Titan Krios electron microscope at either 160,000× (0.67 Å pixel size) or 130,000× (0.64 Å pixel size) on a Gatan K2 or K3 detector equipped with an energy filter. A similar pipeline was used for all datasets (see below). Representative micrographs are provided in Supplementary Figs. 2–14. Contrast transfer function estimation, motion correction and particle picking were done

concurrent to data collection with WarpEM (version 1.0.8)[38]. Good particles (as selected by WarpEM) were imported into CryoSPARC where 2D classification was done. A subselection of particles were made taking the best 2D classes with the highest resolution (below 4 Å for those that led to a high-resolution structure) and largest particle number (classes with more than 5,000 particles) with protein-like features. This particle selection was then used in multiclass ab initio reconstruction. The classes were evaluated and then used as starting references for heterogeneous refinement, this time using all of the good particles from Warp's picking process. The best heterogeneous refinement classes and their particle subsets were then used for homogeneous refinement and in some cases nonuniform refinement in CryoSPARC (Structura Biotechnology; versions 3.0.0 and 3.1.0)[39,40]. The Dis3L2–hairpinA–GCU₁₄ dataset was also processed in Relion using their 3D classification, refinement, contrast transfer function refinement and particle polishing[41–43]. Two representative workflows (processing of the Dis3L2$^{D391N}$–hairpinA–GCU₁₄ and Dis3L2–hairpinD–U₇ datasets) are shown in Supplementary Fig. 1.

To assess the distribution of particles between different Dis3L2 conformations, we used the following standardized protocol for data processing: (1) particles from the datasets were picked using WarpEM's neural network-based picker; (2) good particles were then classified in CryoSPARC's 2D classification; (3) the best 2D classes (as described above) were selected for ab initio reconstruction using five classes (four classes were used for the datasets hairpinD–U₈ #2, hairpinD–U₉ #2 and hairpinD–U₇ #2); and (4) the resulting five ab initio models were used as starting references for heterogeneous refinement using all of the good particles found by the WarpEM picker (Extended Data Fig. 5a). This allowed us to compare the proportion of particles in the full dataset that contributed to a particular Dis3L2 conformation. To ensure that the class distributions were not a result of RNA degradation, control datasets with EDTA were also analyzed and showed the same overall distribution (Extended Data Fig. 5b). To ensure that the 2D selection was not introducing bias into the distribution, we also processed the hairpinC–U₁₂ dataset without 2D classification. In other words, all particles were included in the ab initio and subsequent heterogeneous refinement (Extended Data Fig. 6d,e). Further independent repeat datasets were collected and processed for Dis3L2 complexes with hairpinD–U₅, –U₇, –U₈ and –U₉ (Extended Data Fig. 6c).

### Atomic model building and refinement

Atomic model building and refinement were done in Coot and Phenix (version 1.18-3855-000)[44–46]. Since the mouse and human Dis3L2 proteins are extremely similar in sequence, the mouse Dis3L2 structure (Protein Data Bank (PDB) accession code 4PMW) was used as a starting reference for model building[17]. Once the reference structure was fit into the cryo-EM map, Real-Space Refine was used, with morphing, simulated annealing and rigid body fit in the first rounds[45]. After manual building and correction of geometric outliers and clashes using Coot, further rounds of refinement were done using secondary structure restraints, as well as global minimization, refinement of atomic displacement parameters ($B$ factors) and local grid search. Refinements of complexes with RNA contained further base pair and base stacking restraints in the double-stranded regions. RNA–protein interactions were found with PDBePISA (https://www.ebi.ac.uk/pdbe/pisa/) and examined manually. Final model validation metrics are provided in Table 1. Electrostatics were calculated using PyMol 2.2.3 (Schrödinger) at an ionic strength of 150 mM. All other molecular graphics were performed with UCSF ChimeraX version 0.92, developed by the Resource for Biocomputing, Visualization, and Informatics at the University of California, San Francisco, with support from National Institutes of Health R01-GM129325 and the Office of Cyber Infrastructure and Computational Biology, National Institute of Allergy and Infectious Diseases[47,48].

## Table 1 | Cryo-EM data collection, refinement and validation statistics

| | RNA-free HsDis3L2$^{D391N}$ (EMDB-27827 and PDB 8E27) | HsDis3L2 with hairpinA–GCU$_{14}$, (EMDB-27828 and PDB 8E28) | HsDis3L2 with hairpinC–U$_{12}$ (EMDB-27829 and PDB 8E29) | HsDis3L2 with hairpinD–U$_7$ (EMDB-27830 and PDB 8E2A) |
|---|---|---|---|---|
| **Data collection and processing** | | | | |
| Magnification | 160,000× | 160,000× | 130,000× | 130,000× |
| Voltage (kV) | 300 | 300 | 300 | 300 |
| Electron exposure (e⁻ per Å$^2$ frame) | 2.07 | 2.07 | 2.08 | 2.08 |
| Defocus range (µm) | −2.8 to −0.5 | −2.8 to −0.5 | −2.8 to −0.5 | −2.8 to −0.5 |
| Pixel size (Å) | 0.66 | 0.66 | 0.67 | 0.67 |
| Symmetry imposed | / | / | / | / |
| Initial particle images (number) | 990,287 | 1,256,757 | 1,375,384 | 1,836,786 |
| Final particle images (number) | 162,793 | 557,901 | 531,561 | 539,985 |
| Map resolution (Å) | 3.4 | 3.1 | 3.1 | 2.8 |
| FSC threshold | 0.143 | 0.143 | 0.143 | 0.143 |
| **Refinement** | | | | |
| Initial model used | 4PMW | 4PMW | 4PMW | 4PMW |
| Model resolution | 2.95 | 2.95 | 2.95 | 2.95 |
| Refinement program and final refinement | CryoSPARC and homogeneous refinement | RELION and Refine 3D | CryoSPARC and nonuniform refinement | CryoSPARC and nonuniform refinement |
| FSC (model) 0/0.143/0.5 | | | | |
| Masked | 3.3/3.4/3.6 | 3.0/3.1/3.2 | 3.0/3.2/3.4 | 2.7/2.8/2.9 |
| Unmasked | 3.3/3.4/3.8 | 3.1/3.1/3.3 | 3.0/3.3/3.5 | 2.8/2.8/3.0 |
| Map sharpening B factor (Å$^2$) | −150.7 | −79 | −160.4 | −109.8 |
| Model composition | | | | |
| Atoms | 5,371 (H: 0) | 5,789 (H: 0) | 6,197 (H: 0) | 5,768 (H: 0) |
| Residues | Protein: 676 Nucleotide: 0 | Protein: 690 Nucleotide: 15 | Protein: 692 Nucleotide: 33 | Protein: 686 Nucleotide: 15 |
| B factors (Å$^2$) | | | | |
| Protein | 29.11/136.95/79.07 | 16.07/113.65/45.17 | 37.96/150.47/75.76 | 39.39/141.89/71.65 |
| Nucleotide | / | 38.44/86.94/55.20 | 60.65/210.35/160.78 | 51.96/143.50/104.02 |
| RMS deviations | | | | |
| Bond lengths (Å) | 0.012 (0) | 0.004 (0) | 0.012 (0) | 0.004 (0) |
| Bond angles (°) | 1.173 (1) | 0.827 (0) | 1.152 (0) | 0.699 (0) |
| **Validation** | | | | |
| MolProbity score | 1.06 | 1.17 | 1.31 | 1.44 |
| Clash score | 2.69 | 3.83 | 5.62 | 6.65 |
| Poor rotamers (%) | 0.34 | 0 | 0.66 | 0 |
| Ramachandran plot | | | | |
| Disallowed (%) | 0 | 0 | 0 | 0 |
| Allowed (%) | 1.20 | 1.46 | 1.46 | 2.35 |
| Favored (%) | 98.8 | 98.54 | 98.54 | 97.65 |

FSC, Fourier shell correlation; RMS, root mean square.

### Presteady-state and quasisteady-state nuclease reactions

Nuclease reactions were performed in a temperature-controlled heat block at 20 °C in a total volume of 40 µl. Reaction mixtures containing 20 mM HEPES (pH 7.0), 50 mM NaCl, 5% glycerol, 100 µM MgCl$_2$, 1 mM DTT and Dis3L2 were preincubated for 5 min. Presteady-state reactions were started by the addition of 5′ radiolabeled RNA substrate to a final concentration of 1 nM. The concentrations of Dis3L2 were in far excess of the RNA and ranged from 5–1,000 nM, as indicated. Measurements were taken at the time points 7 s, 15 s, 30 s, 1 min, 2 min, 3 min, 5 min, 10 min and 15 min, except for long-time-course experiments for which the times are indicated (Extended Data Fig. 7b,c). Reactions were quenched by the addition to an equal volume of stop buffer (80% formamide, 0.1% bromophenol blue, 0.1% xylene cyanole, 2 mM EDTA and 1.5 M urea). Samples were heated to 95 °C and analyzed on sequencing

gels composed of 20% acrylamide and 7 M urea. Gels were exposed to phosphor screens overnight and scanned with a Typhoon FLA 7000 imager (GE Healthcare Life Sciences). Bands were quantified using SAFA footprinting software and the values were normalized for each lane[49]. For a typical reaction with a 34-nucleotide substrate and ten time points, we quantified all species larger than the five-nucleotide end product and obtained approximately 300 data points for each Dis3L2 concentration.

### Pulse–chase nuclease reactions

Pulse–chase reactions were performed under conditions identical to those for presteady-state reactions. Reactions were initiated by the addition of enzyme and allowed to proceed for a defined period of time ($t_1$). At $t_1$, an excess of cold scavenger RNA (×5,000-fold) was added to a final concentration of 5 µM. After incubation for the indicated time ($t_2$), aliquots were removed and quenched in stop buffer. Samples were analyzed on sequencing gels and processed as for the above-described presteady-state reactions.

### Calculation of kinetic parameters

Kinetic parameters were obtained using a global fit of the data from presteady-state titrations and pulse–chase experiments. Global data fitting was performed using the Kinetic Explorer software (version 8.0; KinTek Global)[50,51]. Initial parameters for the global fit were: observed rate constants ($k_{obs}$), processivity values ($P$) and $K_{1/2}$ and $k_{obs}^{max}$ values for each reaction species. Observed rate constants ($k_{obs}$) were calculated from presteady-state experiments by fitting each experiment separately using the global data-fitting software GFIT[52] to a model that calculates rate constants for a series of irreversible, pseudo-first-order reactions. Initial parameters for GFIT were obtained by fitting the disappearance of a 34-nucleotide substrate to a first-order exponential: $y = a_1 \times \exp(-b_1 \times t) + c$, where $a_1$ is the amplitude, $b_1$ is the observed rate constant ($k_{obs}$) and $c$ is the offset. Processivity values for individual degradation steps were determined from the distribution of substrate species before and after scavenger addition[30] (Extended Data Fig. 7g), where processivity ($P$) was defined as: $P = k_f / (k_f + k_{off})$.

The equations to calculate processivity values from distributions of species were fit using a customized script in the Mathematica software package (Wolfram)[30]. To derive the $K_{1/2}$ and $k_{obs}^{max}$ values, we fit $k_{obs}$ versus Dis3L2 concentration data to a binding isotherm function defined as: $k_{obs} = (k_{obs}^{max} \times [\text{Dis3L2}]) \times (K_{1/2}^{\text{Dis3L2}} + [\text{Dis3L2}])^{-1}$. $K_{1/2}$ is the functional equilibrium dissociation constant and $k_{obs}^{max}$ is the maximal observed rate constant at enzyme saturation (Extended Data Fig. 7d). The data were then evaluated by plotting in GraphPad Prism version 9.1.2 (GraphPad Software). These initial parameters were used as guides in setting up a range of starting values for the elementary rate constants in a global fit to the minimal kinetic model, as shown in Extended Data Fig. 7a. The $K_{1/2}$ values were used to constrain the ratio of dissociation and association rate constants for productive binding by linking the two values as initial parameters. The $k_{obs}^{max}$ values were used to set boundaries on the forward rate constant, $k_f$. Finally, the experimentally determined processivity values ($P$) were used as initial constraints on the ratio of $k_f$ and $k_{off}$. The global data fit was done in an iterative manner by alternating combinations of fixed and floating variables while tracking the overall $\chi^2$ value. The goodness of the fit, $R^2 = 0.94$, was calculated by plotting the experimental datasets versus the corresponding simulated data from the kinetic model (Extended Data Fig. 7f). As an additional measure of the overall quality of fit, we performed FitSpace analysis[51] to determine the lower and upper boundaries of each kinetic parameter (Supplementary Tables 1–3). For a typical substrate, roughly 2,500 individual data points from enzyme titrations and 750 data points from pulse–chase experiments were used to calculate the 120 kinetic parameters that describe degradation of a 34-nucleotide substrate down to five nucleotides.

Errors for elementary rate constants represent standard errors of the mean from the global data fitting. Errors for compound rate constants, such as processivity and $K_{1/2}$, were calculated via the error propagation formulas shown below.

$$\sigma_{K_{1/2}} = K_{1/2} \sqrt{\frac{\sigma_{k_{off}}^2}{k_{off}^2} + \frac{\sigma_{k_{on}}^2}{k_{on}^2}}$$

$$\sigma_P = P(1 - P) \sqrt{\frac{\sigma_{k_{off}}^2}{k_{off}^2} + \frac{\sigma_{k_f}^2}{k_f^2}}$$

### Reporting summary

Further information on research design is available in the Nature Portfolio Reporting Summary linked to this article.

## Data availability

Structure coordinates and cryo-EM data have been deposited in the PDB and Electron Microscopy Data Bank (EMDB), respectively. The structures can be found under the following accession numbers: PDB 8E27 and EMDB-27827 (RNA-free HsDis3L2); PDB 8E28 and EMDB-27828 (HsDis3L2 in complex with hairpinA–GCU₁₄); PDB 8E29 and EMDB-27829 (HsDis3L2 in complex with hairpinC–U₁₂); and PDB 8E2A and EMDB-27830 (HsDis3L2 in complex with hairpinD–U₇). The cryo-EM map of the low-resolution HsDis3L2 complex with hairpinE–U₇ has been deposited in the EMDB under accession code EMDB-27831. The structure of mouse Dis3L2 (PDB 4PMW) was used as a reference and for comparisons. Source data are provided with this paper.

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

## Acknowledgements

We thank D. Herschlag for detailed discussions, comments and recommendations, J. Kinney for advice, S. Jones and J. Scheuring for technical support, J. Ipsaro for comments on the manuscript, M. Jaremko for technical advice and other members of the L.J. laboratory for helpful suggestions. Cryo-EM was performed at the Cold Spring Harbor Laboratory (CSHL) cryo-EM facility. We also thank the CSHL Mass Spectrometry Shared Resource, which is supported by the Cancer Center Support Grant 5P30CA045508. This work was supported by NIH grant R01-GM114147 (to L.J.) and the CSHL School of Biological Sciences (to K.M., A.D. and L.J.). K.M. was supported by the Leslie C. Quick, Jr. Fellowship. L.J. is an investigator of the Howard Hughes Medical Institute.

## Author contributions

K.M. and L.J. designed the study. K.M. purified the proteins used in this work and preformed all of the cryo-EM sample preparation, data collection, data processing and model building. D.R.T. assisted in cryo-EM data collection. A.D. contributed to the purification and cryo-EM analysis of RNA-free and hairpinA–GCU$_{14}$ HsDis3L2. A.A. designed and led the kinetic analysis of HsDis3L2. A.A. and K.M. carried out the kinetic assays and data analysis. K.M., A.A. and L.J. prepared the manuscript.

## Competing interests

The authors declare no competing interests.

## Additional information

**Extended data** is available for this paper at https://doi.org/10.1038/s41594-023-00923-x.

**Correspondence and requests for materials** should be addressed to Leemor Joshua-Tor.

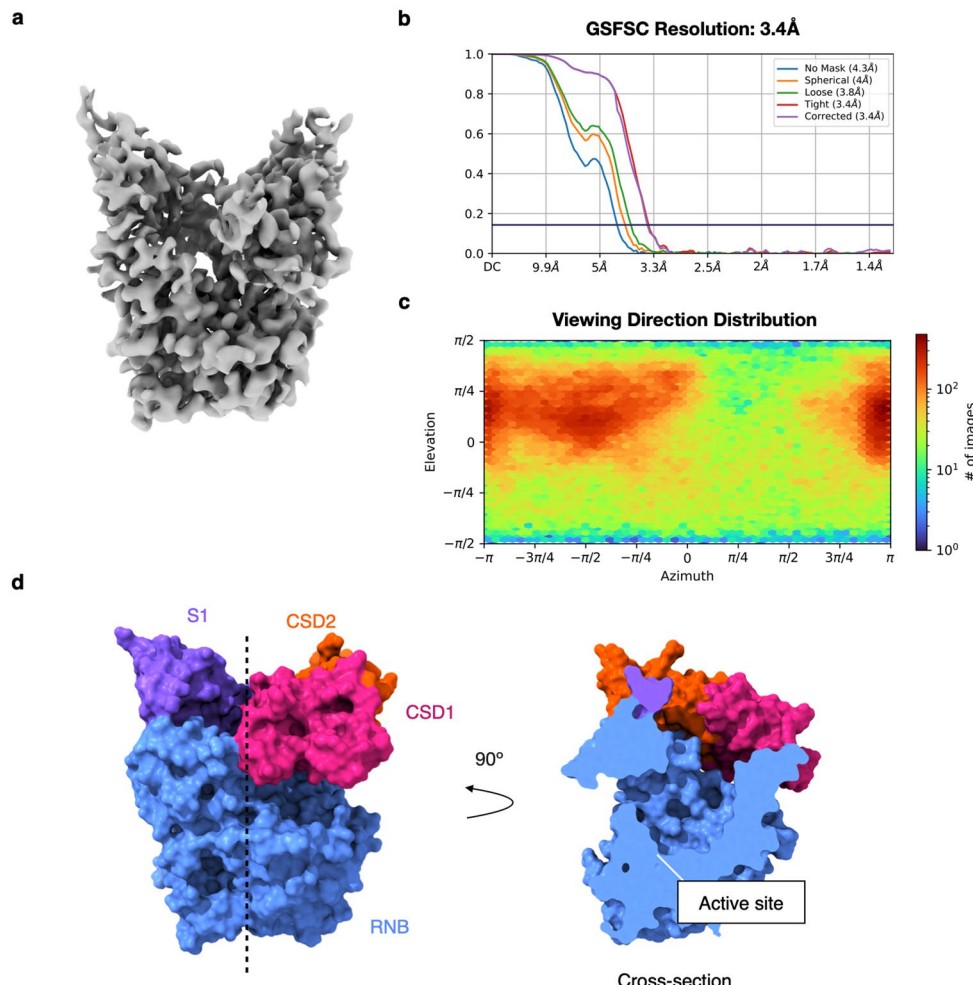

**Extended Data Fig. 1 | Cryo-EM structure of RNA-free human Dis3L2. a**, Final 3D map of human Dis3L2 (left) with **b**, Gold Standard (0.143) Fourier Shell Correlation (GSFSC) and **c**, viewing direction distribution (bottom right). The 3D map was produced by homogeneous refinement of roughly 163-thousand particles in CryoSPARC. **d**, Surface representation and cut-away (right) showing the tunnel and the active site.

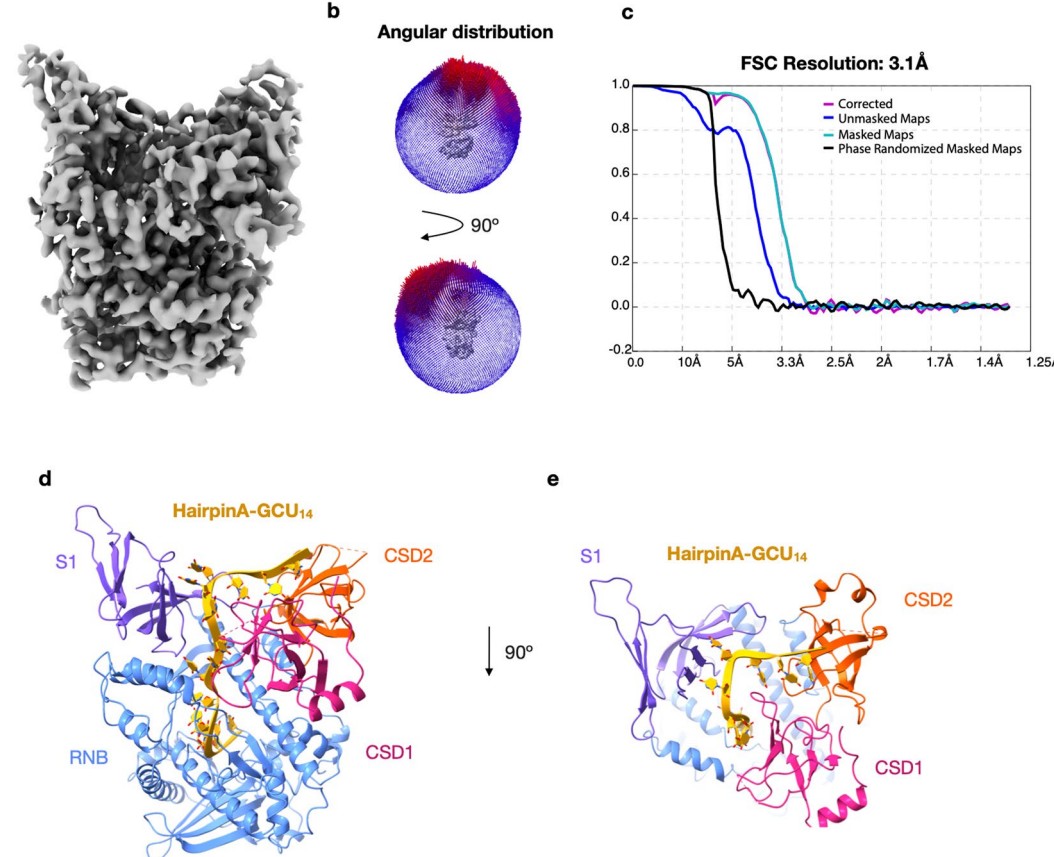

**Extended Data Fig. 2 | Initial substrate binding by human Dis3L2. a**, Final 3D map of human Dis3L2 in complex with hairpinA-GCU$_{14}$ with **b**, particle angular distribution and **c**, GSFSC. The final 3D map was produced by 3D refinement of roughly 101,000 particles in Relion. **d**, Structure of Dis3L2 in complex with hairpinA-GCU$_{14}$ with domains indicated, **e**, and view of top/apical face (rotated 90° toward the reader).

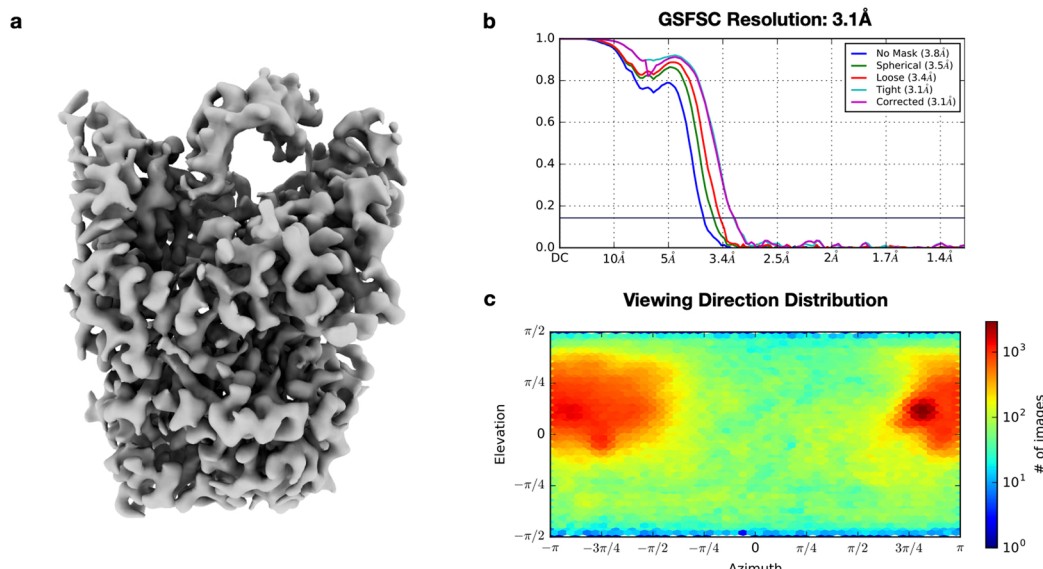

**Extended Data Fig. 3 | Engaging double-stranded RNA by Dis3L2. a**, Final 3D map of human Dis3L2 in complex with hairpinC-U$_{12}$ with **b**, FSC and **c**, the particle viewing direction distribution. The final 3D map was produced by homogeneous refinement of roughly 532,000 particles in cryoSPARC.

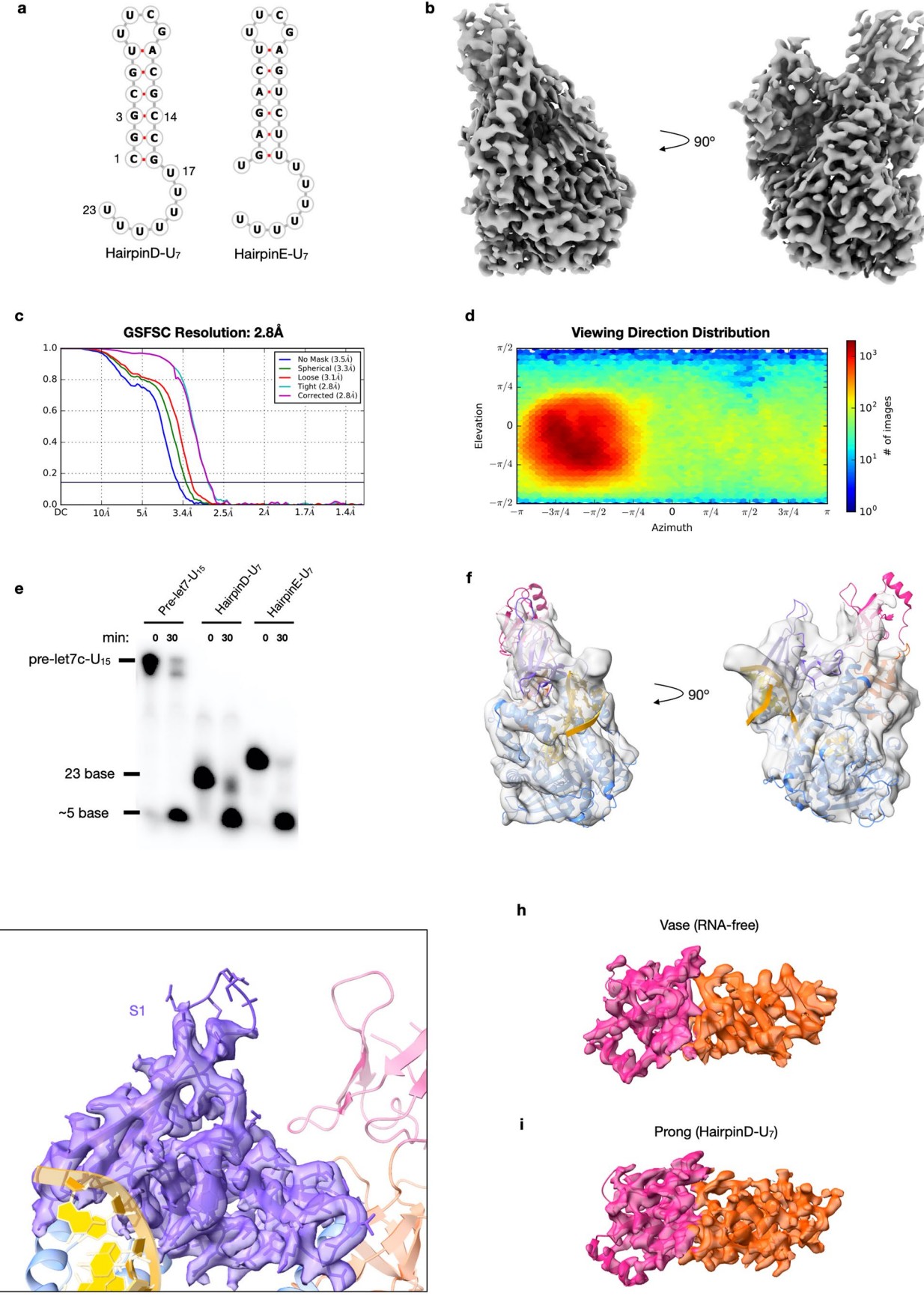

**Extended Data Fig. 4 | See next page for caption.**

**Extended Data Fig. 4 | Conformational change is triggered by structured RNA substrates with shortened 3′ overhangs. a**, Predicted RNA-fold for hairpinD-U$_7$ (75%- GC content of dsRNA) and hairpinE-U$_7$ (50% GC content of dsRNA). **b**, Final 3D map of human Dis3L2 in complex with hairpinD-U7 superimposed with the structure of RNA-free Dis3L2 fit in the map. **c**, GSFSC and **d**, particle viewing direction distribution. The final 3D map was produced by homogeneous refinement of roughly 539,000 particles in cryoSPARC. **e**, Urea-PAGE of nuclease activity assay using pre-let7-U$_{15}$, hairpinD-U$_7$ and hairpinE-U$_7$ as substrates. Species length is indicated on the left of the gel. **f**, Unsharpened map of Dis3L2 in complex with the 50% GC-rich hairpinE-U$_7$ (homogeneous refinement with 226,117 particles) with structure of Dis3L2-hairpinD-U$_7$ fit in the map. **g**, Density overlay of the S1 domain. **h**, Density for the CSDs in the vase conformation **i**, and in the prong.

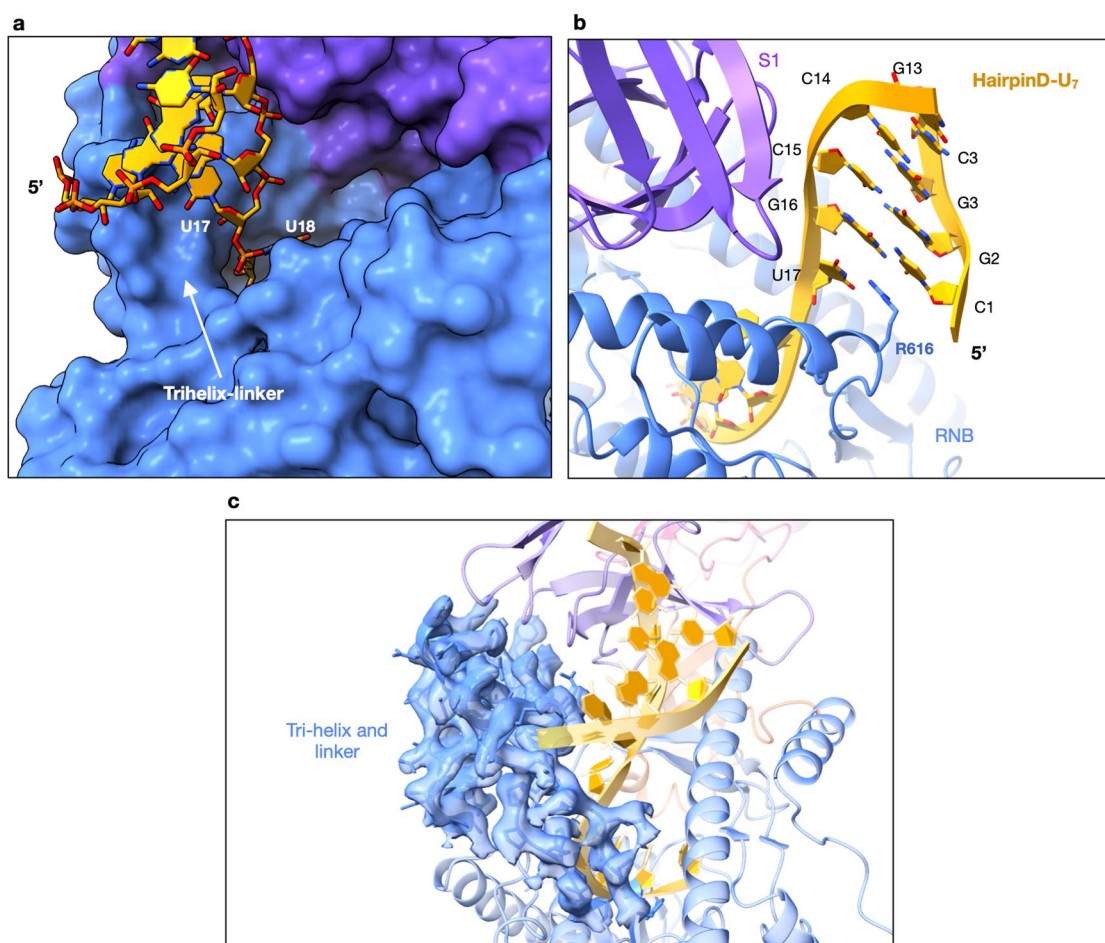

**Extended Data Fig. 5 | The base of the double-helical stem is positioned above an RNB trihelix-linker. a**, The trihelix-linker forms the final barrier to the dsRNA before the tunnel to the active site. **b**, R616 on the junction of the trihelix stacks under C1 and points towards the first base of the 3'overhang (U17). **c**, Density for the RNB trihelix and linker.

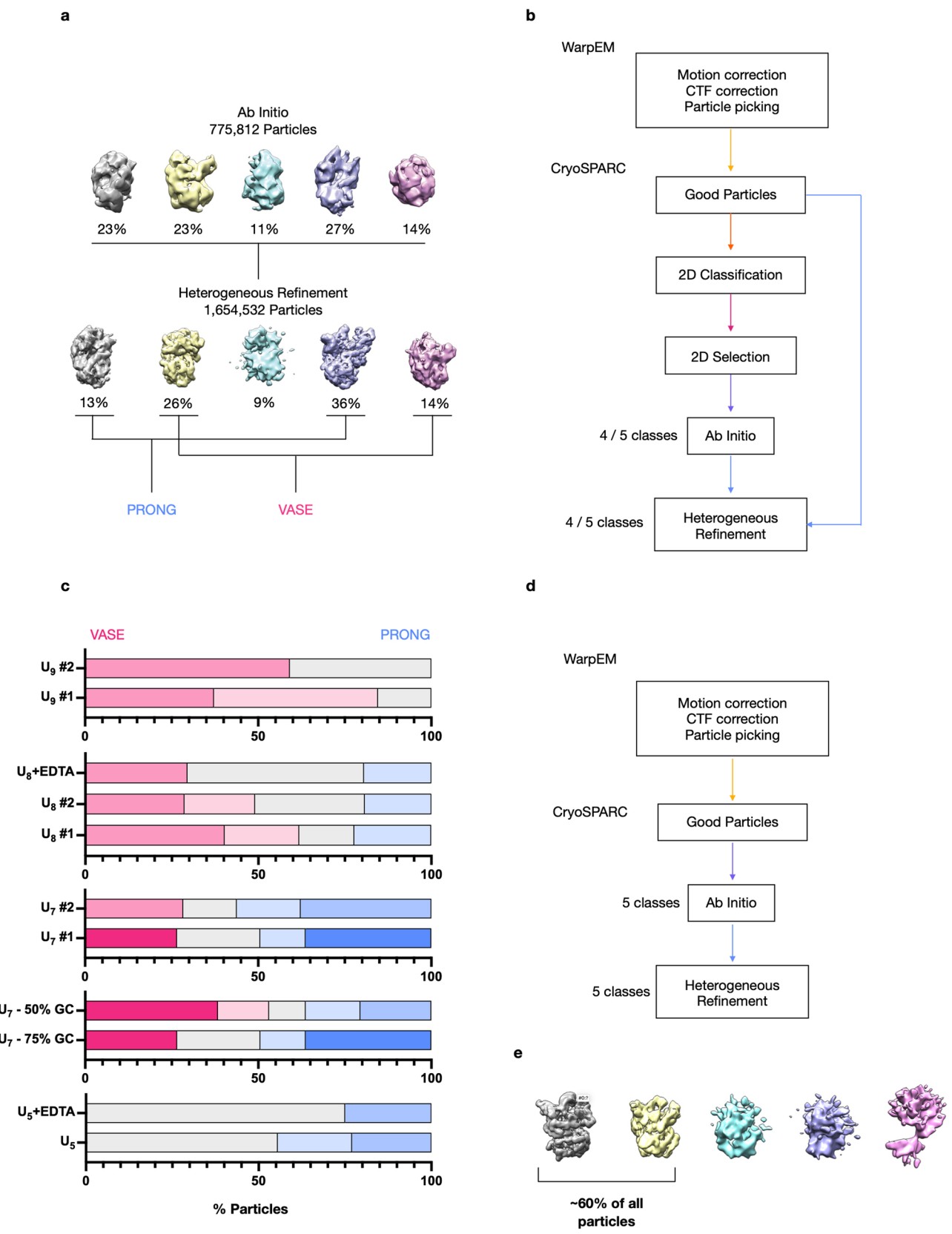

Extended Data Fig. 6 | See next page for caption.

**Extended Data Fig. 6 | CryoEM 3D class distribution of particles.**
**a**, Representative processing pipeline of hairpinD-$U_7$ showing the presence of both vase and prong conformation. **b**, Standardized data processing workflow to compare 3D class distribution of particles (Methods). Good particles picked by WarpEM's neural network-based picker first underwent 2D classification in cryoSPARC. Particles from classes with high estimated resolution and particle number were selected for ab initio reconstruction with 4 to 5 classes (Methods). The resulting initial models were then used as starting references in a heterogeneous refinement with the full set of good particles. The resulting refinements were evaluated for their similarity to the prong or vase conformation and the proportion of particles belonging to each class plotted in **c**.

**c**, Comparison of particle distribution in different datasets. Numbers #1 and #2 indicate repeat data collections of the same complex. Controls with EDTA (100 μM) were done for the hairpinD-$U_8$ and -$U_5$ complexes. Effect of hairpin GC content on conformational distribution of Dis3L2 was evaluated for a -$U_7$ overhang. X-axis: % of particles in vase (pink) or prong (blue) conformation, y-axis: individual datasets RNA-free or hairpin RNA-bound Dis3L2, numbers denote the length of the 3′ overhang. The deeper color indicates higher quality 3D-reconstructions, gray indicates particles that did not contribute to a meaningful reconstruction. **d**, Control processing pipeline excluding 2D classification. **e**, 3D maps from heterogeneous refinement of hairpinC-$U_{12}$ dataset in CryoSPARC with the proportion of particles in a vase conformation indicated.

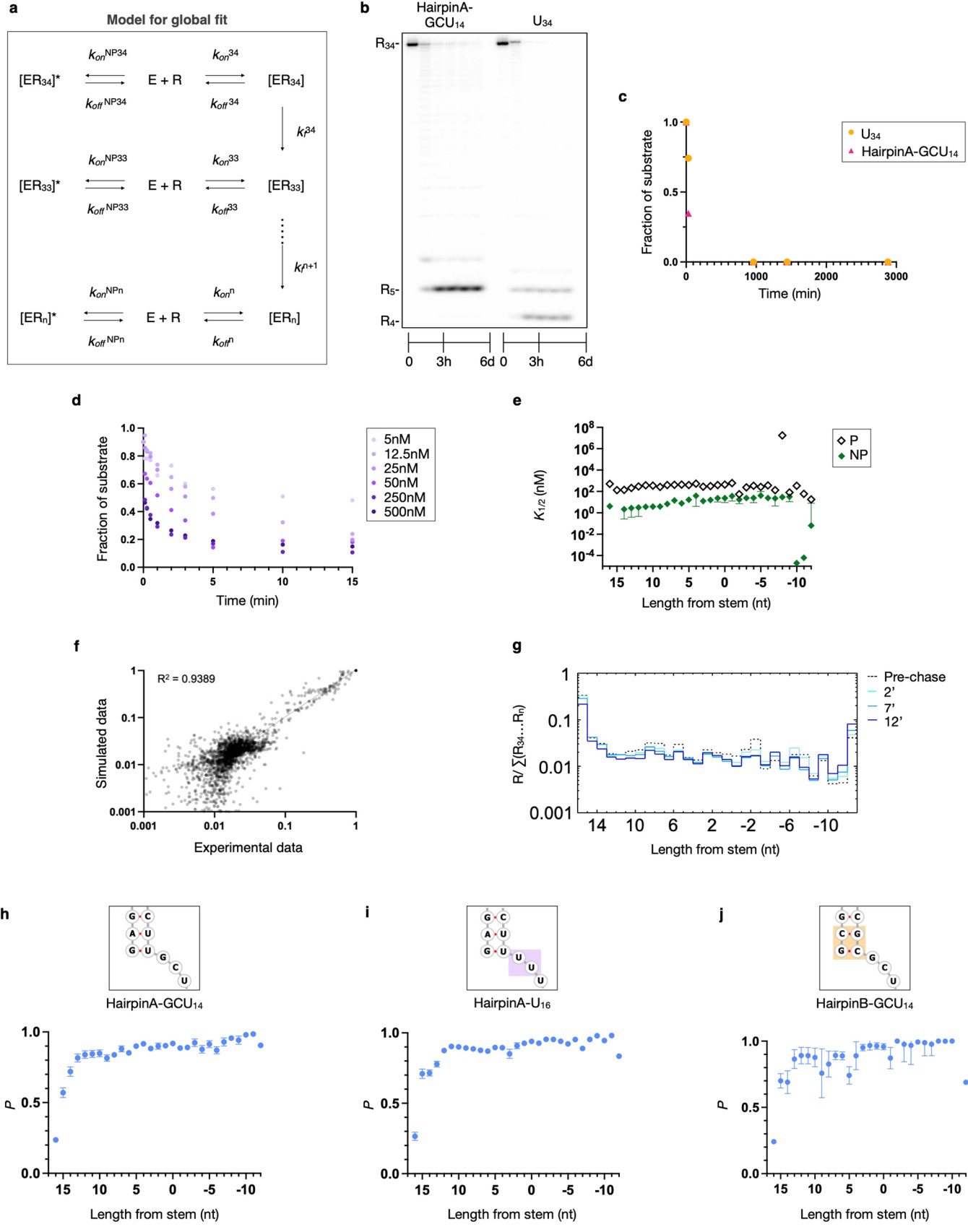

**Extended Data Fig. 7 | See next page for caption.**

**Extended Data Fig. 7 | Kinetic analysis of WT HsDis3L2 against hairpin RNAs. a**, Minimal kinetic model used for global data fitting. **b**, Gel showing degradation of hairpinA-GCU$_{14}$ and ss-U$_{34}$ to completion by Dis3L2 over a longer time course. **c**, Quantification of substrate disappearance from reaction shown in **b**. **d**, Fraction of hairpinA-GCU$_{14}$ substrate disappearance measured over time at various Dis3L2 concentrations. **e**, Comparison of functional equilibrium dissociation constants for productive ($K_{1/2}{}^{P}$) and non-productive ($K_{1/2}{}^{NP}$) binding determined in the global fit of WT Dis3L2 on hairpinA-GCU$_{14}$ (the data point for x = 15 was omitted due to large uncertainty). **f**, Correlation of experimental data vs the corresponding data calculated using the kinetic parameters for pre-steady state reactions of Dis3L2 on hairpinA-GCU$_{14}$. **g**, Step-plot showing

the change in substrate and intermediate species at discrete timepoints during pulse-chase reaction at 50 nM Dis3L2 on hairpinA-GCU$_{14}$, the initial timepoint was taken before addition of chase at 3 min. **h-j**, Comparison of Dis3L2 processivity on hairpinA-GCU$_{14}$, hairpinA-U$_{16}$, and hairpinB-GCU$_{14}$ (P$^{\text{HairpinB-GCU14}}$, x = -11 was omitted due to large uncertainty). Error bars are shown as vertical lines. $K_{1/2}$ errors in (e) were calculated using error propagation from SEM of $k_{off}$ and $k_{on}$ derived from global fit of data from enzyme titrations (9 concentrations with n = 5) and pulse chase experiments (2 concentrations with n = 4). Processivity error for **i** represents the Standard Deviation of the Mean with n = 3, while errors for **h** and **j** were calculated using the error propagation formula from the SEM of $k_f$ and $k_{off}$ as described above for panel e.

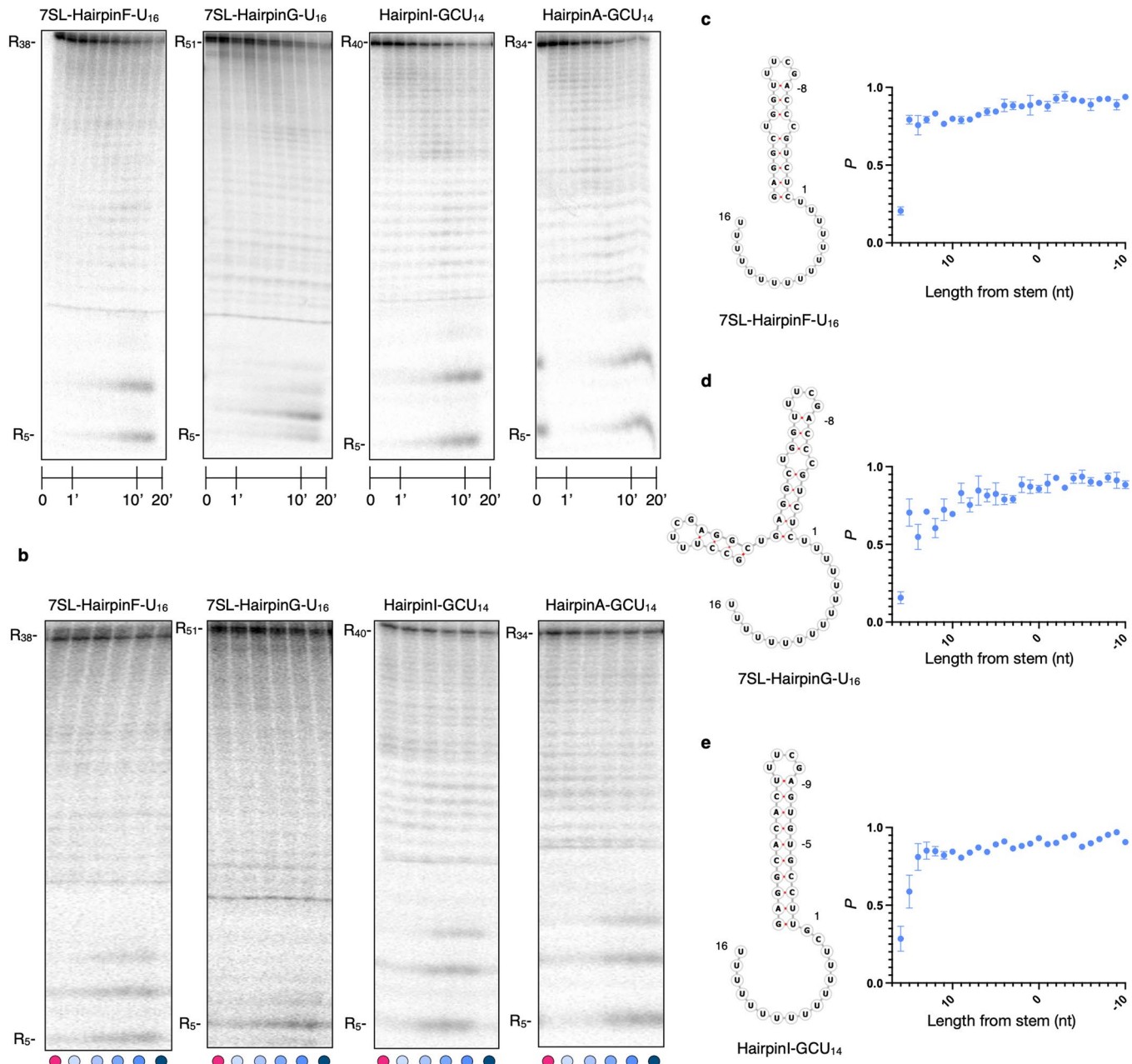

**Extended Data Fig. 8 | Processivity of HsDis3L2 on 7SL mimic and longer double-stranded RNA substrates. a**, Representative gels from pre-steady state nuclease titration assays with 1 nM 5′ P³² -radiolabeled RNA substrate and 50 nM HsDis3L2. The overall length of the species is indicated on the left of the panels. Each experiment was repeated independently with similar results for three concentrations of HsDis3L2. **b**, Representative gels from pulse-chase reactions of WT human Dis3L2 at 50 nM concentration and 1 nM radiolabeled

RNA substrate. Cold chase was added to the reaction at the 3 min timepoint to a final concentration 5000 nM. Measurements were taken pre-chase at 3 min (pink dot), and post-chase at 4, 5, 7.5, and 10 min (blue gradient dots). Each experiment was repeated independently with similar results at two concentrations of HsDis3L2 with n = 2. **c, d, e**, RNA fold predictions and corresponding processivity values from pulse-chase experiments for 7SL-hairpinF-U₁₆, 7SL-hairpinG-U₁₆, and hairpinI-GCU₁₄. Error bars mark the Standard Deviation of the Mean with n = 4.

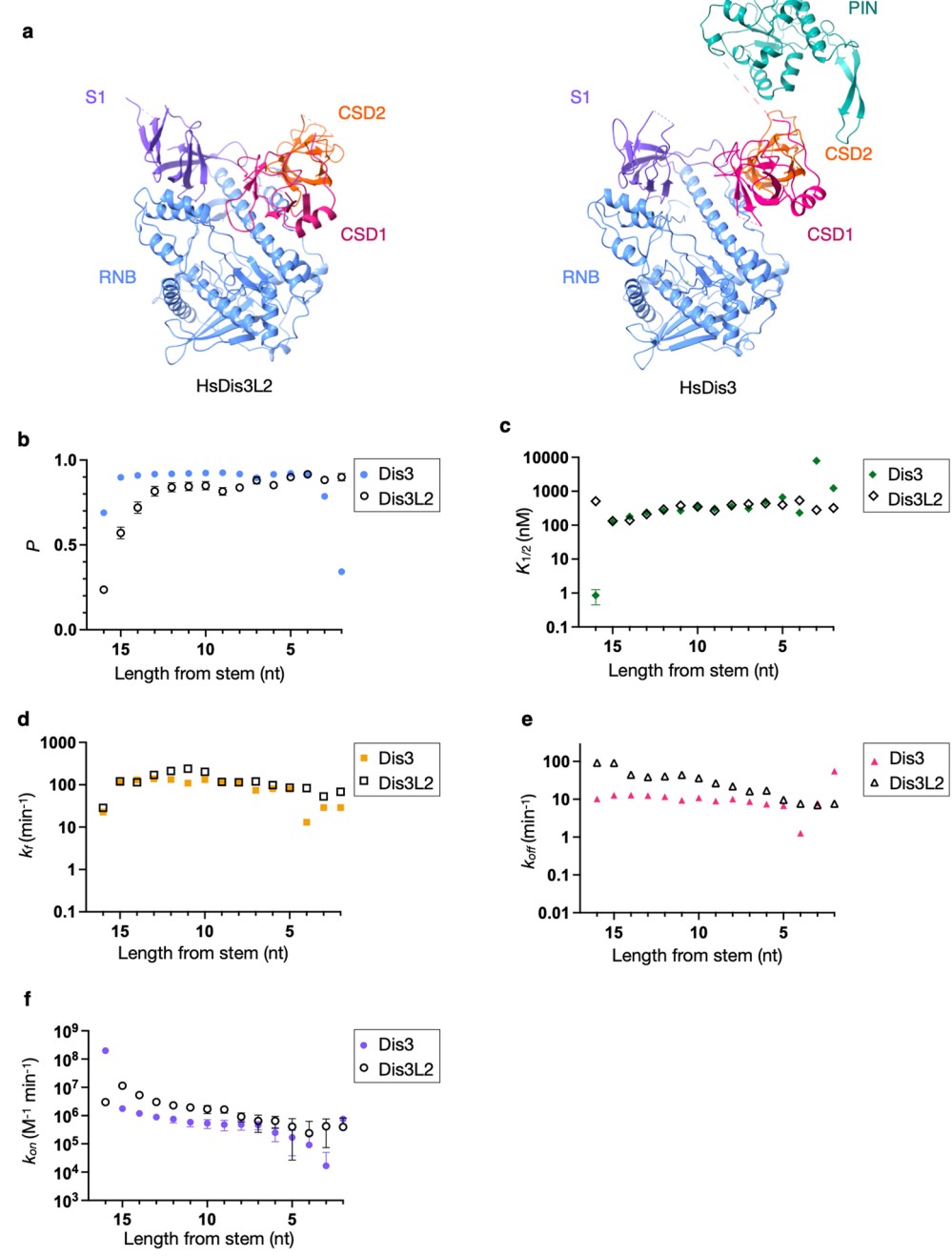

**Extended Data Fig. 9 | Comparison of structures and kinetic profiles of human Dis3 and Dis3L2. a**, Structures of Human Dis3L2 and Human Dis3 (PDB code: 6D6Q)[51]. **b**–**f**, Kinetic profile for hairpinA-GCU$_{14}$ degradation by HsDis3 at single-nt resolution: **b**, Processivity (blue circles), **c**, Functional equilibrium dissociation constant for productive binding (K$_{1/2}$) (green diamonds), **d**, forward rate constant (orange squares), **e**, dissociation rate constant (pink triangles), and, **f**, association rate constants (purple circles) compared to HsDis3L2 (gray). Error bars for (**d**-**f**) represent the SEM derived from a global fit of data from enzyme titrations (9 Dis3 concentrations with n = 2) and pulse-chase experiments (2 Dis3 concentrations with n = 3). Processivity error bars for (**b**) are propagated errors calculated from the SEM *of* k$_f$ *and* k$_{off}$, *while* K$_{1/2}$ errors (**c**) are propagated errors calculated from SEM *of* k$_{off}$ *and* k$_{on}$.

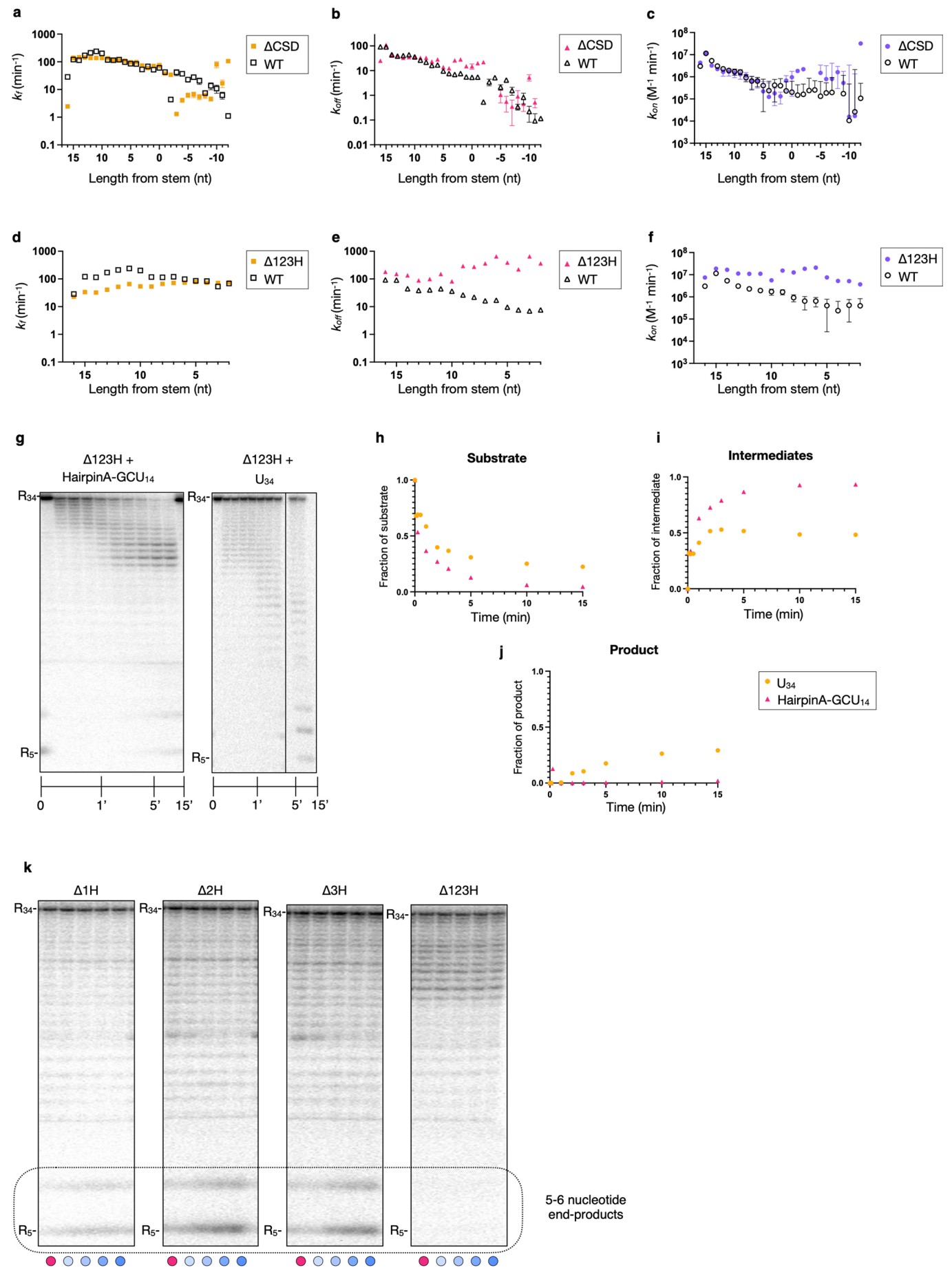

**Extended Data Fig. 10 | See next page for caption.**

**Extended Data Fig. 10 | Contribution of the CSDs and trihelix-and-linker to degradation of structured substrates by human Dis3L2. a–c**, Kinetic profile for hairpinA-GCU$_{14}$ degradation by the ΔCSD mutant at single-nt resolution: **a**, forward rate constants (orange squares), **b**, dissociation rate constants (pink triangles), and, **c**, association rate constants (purple circles) compared to WT Dis3L2 (gray), (the datapoints for k$_{on}$ at x = −3 and −4 were omitted due to large uncertainty). Error bars mark the SEM derived from a global fit of data from enzyme titrations (8 ΔCSD concentrations with n = 2) and pulse-chase experiments (2 ΔCSD concentrations with n = 3). **d–f**, Kinetic profile for hairpinA-GCU$_{14}$ degradation by Δ123H (trihelix deletion) mutant at single-nt resolution: **d**, forward rate constants (orange squares), **e**, dissociation rate constants (pink triangles), and, **f**, association rate constants (purple circles) compared to WT Dis3L2. Error bars mark the SEM derived from a global fit of data from enzyme titrations (9 Δ123H concentrations with n = 3) and pulse-chase experiments (2 Δ123H concentrations with n = 4). **g**, Representative gels showing the degradation profile by 250 nM Δ123H mutant against 1 nM hairpinA-GCU$_{14}$ and U$_{34}$ substrates. Each experiment was repeated independently with similar results for n = 2 **h–j**, Fraction of substrate disappearance, fraction of intermediates, and fraction of end-product accumulation over time from reactions shown in **g**, **k**, Representative pulse-chase experiments for 1 nM hairpinA-GCU$_{14}$ and 100 nM Dis3L2 helix mutants: Δ1H (Δ612-634), Δ2H (Δ632-647), Δ3H (Δ654-665) and Δ123H. Measurements were taken pre-chase at 5 min (pink dot), and post-chase at 6, 7, 10, and 15 min (blue gradient dots). Each experiment was repeated independently with similar results for n = 2.

# Reporting Summary

## Statistics

For all statistical analyses, confirm that the following items are present in the figure legend, table legend, main text, or Methods section.

| n/a | Confirmed | |
|---|---|---|
| ☐ | ☒ | The exact sample size (*n*) for each experimental group/condition, given as a discrete number and unit of measurement |
| ☐ | ☒ | A statement on whether measurements were taken from distinct samples or whether the same sample was measured repeatedly |
| ☒ | ☐ | The statistical test(s) used AND whether they are one- or two-sided<br>*Only common tests should be described solely by name; describe more complex techniques in the Methods section.* |
| ☒ | ☐ | A description of all covariates tested |
| ☐ | ☒ | A description of any assumptions or corrections, such as tests of normality and adjustment for multiple comparisons |
| ☐ | ☒ | A full description of the statistical parameters including central tendency (e.g. means) or other basic estimates (e.g. regression coefficient) AND variation (e.g. standard deviation) or associated estimates of uncertainty (e.g. confidence intervals) |
| ☒ | ☐ | For null hypothesis testing, the test statistic (e.g. *F*, *t*, *r*) with confidence intervals, effect sizes, degrees of freedom and *P* value noted<br>*Give P values as exact values whenever suitable.* |
| ☒ | ☐ | For Bayesian analysis, information on the choice of priors and Markov chain Monte Carlo settings |
| ☒ | ☐ | For hierarchical and complex designs, identification of the appropriate level for tests and full reporting of outcomes |
| ☒ | ☐ | Estimates of effect sizes (e.g. Cohen's *d*, Pearson's *r*), indicating how they were calculated |

*Our web collection on statistics for biologists contains articles on many of the points above.*

## Software and code

Policy information about availability of computer code

| | |
|---|---|
| Data collection | EPU ver 2.7 (ThermoFisher) |
| Data analysis | cryoEM: WarpEM ver 1.0.6 & 1.0.9 (Tegunov & Cramer); cryoSPARC v3.0.0 and v3.1.0 (Structura Biotechnology); Relion v 3.0 & 3.1 (Zivanov et al 2018) ; Coot 0.8.9.1 (Emsley & Cowtan 2004); Phenix 1.18-3855 (Afonine et al 2018, Terwilliger et al, 2020); Pymol 2.2.3 (Schrodinger, LLC); ChimeraX, ver 0.92 (RBVI);<br>kinetics: SAFA 1.0 (Das et al, 2005); Kinetic Explorer ver 8.0 (Kintek Global); GFIT 1.0 (Levin et al, 2009); Mathematica ver 6.0 (Wolfram); GraphPad Prism ver 9.1.2 (GraphPad Software).<br>other: FoRNA v1.0; Benchling (no version number, used 2019-2022); RNAfold (no version number). |

For manuscripts utilizing custom algorithms or software that are central to the research but not yet described in published literature, software must be made available to editors and reviewers. We strongly encourage code deposition in a community repository (e.g. GitHub). See the Nature Portfolio guidelines for submitting code & software for further information.

# Data

Policy information about availability of data

All manuscripts must include a data availability statement. This statement should provide the following information, where applicable:

- Accession codes, unique identifiers, or web links for publicly available datasets
- A description of any restrictions on data availability
- For clinical datasets or third party data, please ensure that the statement adheres to our policy

Structure coordinates and cryoEM data have been deposited in the PDB and EMDB respectively. The structures can be found under the following PDB accession numbers: RNA-free HsDis3L2: PDB 8E27, EMDB-27827; HsDis3L2 in complex with hairpinA-GCU14: PDB 8E28, EMDB-27828; HsDis3L2 in complex with hairpinC-U12: PDB 8E29, EMDB-27829; HsDis3L2 in complex with hairpinD-U17: PDB 8E2A, EMDB-27830. The cryoEM map of the low resolution HsDis3L2 complex with hairpinE-U7 was deposited in the EMDB under accession code EMDB-27831. The structure of mouse Dis3L2 (PDB 4PMW) was used as a reference and for comparisons. Source data for the kinetic analysis are provided with this paper.

# Human research participants

Policy information about studies involving human research participants and Sex and Gender in Research.

| Reporting on sex and gender | n/a |
|---|---|
| Population characteristics | n/a |
| Recruitment | n/a |
| Ethics oversight | n/a |

Note that full information on the approval of the study protocol must also be provided in the manuscript.

# Field-specific reporting

Please select the one below that is the best fit for your research. If you are not sure, read the appropriate sections before making your selection.

☒ Life sciences  ☐ Behavioural & social sciences  ☐ Ecological, evolutionary & environmental sciences

For a reference copy of the document with all sections, see nature.com/documents/nr-reporting-summary-flat.pdf

# Life sciences study design

All studies must disclose on these points even when the disclosure is negative.

| Sample size | For pulse-chase experiments, at least three replicate reactions were run at two different enzyme concentrations and the data were fit separately to analytical functions to calculate mean +/- standard deviation. For enzyme titration experiments, we tested at least five different enzyme concentrations with each reaction/concentration performed multiple times as indicated in the attached table.<br>No statistical methods were used to determine sample size. Instead, an appropriate sample size for enzyme titrations and pulse-chase experiments was picked to ensure a large enough dataset to obtain a global fit that yielded kinetic parameters that were well constrained by the data. Goodness of fit and quality of kinetic parameters were judged by examining the Std Error of Mean values and asymmetric error of the parameters via FitSpace analysis. All these values were well below the statistical thresholds that are customary in enzyme kinetic analysis. |
|---|---|
| Data exclusions | see Supplementary Table 4, which includes all requested information. Exclusions were made for kinetic data only due to large uncertainty |
| Replication | Kinetic experiments had a high degree of reproducibility as judged by: i) statistical parameters of mean +/- standard deviation for the analytical fits, ii) best fit parameters +/- standard errors of the mean for the global fits, and iii) asymmetric error analysis of kinetic parameters using the FitSpace feature of Kintek Explorer software.<br>All attempts of replication were successful. |
| Randomization | For the kinetic and structural analyses, there is no opportunity for randomization. |
| Blinding | For particle picking in the cryoEM experiments to determine structures, these were done using criteria in a non-biased way, with no subjective input.<br>Other than that there is no room for blinding in any of the experiments described, no clinical data are analyzed. |

# Reporting for specific materials, systems and methods

We require information from authors about some types of materials, experimental systems and methods used in many studies. Here, indicate whether each material, system or method listed is relevant to your study. If you are not sure if a list item applies to your research, read the appropriate section before selecting a response.

## Materials & experimental systems

| n/a | Involved in the study |
|-----|-----------------------|
| ☒ | ☐ Antibodies |
| ☐ | ☒ Eukaryotic cell lines |
| ☒ | ☐ Palaeontology and archaeology |
| ☒ | ☐ Animals and other organisms |
| ☒ | ☐ Clinical data |
| ☒ | ☐ Dual use research of concern |

## Methods

| n/a | Involved in the study |
|-----|-----------------------|
| ☒ | ☐ ChIP-seq |
| ☒ | ☐ Flow cytometry |
| ☒ | ☐ MRI-based neuroimaging |

## Eukaryotic cell lines

Policy information about cell lines and Sex and Gender in Research

| | |
|---|---|
| Cell line source(s) | Spodoptera frugiperda: pupal ovarian cells (Sf9), Gibco / ThermoFisher Scientific<br>Catalogue Number: 11496015; RRID: CVCL_0549 |
| Authentication | Cell line was not authenticated, it was used solely for the purpose of producing protein. The protein was validated using Mass Spectrometry. |
| Mycoplasma contamination | Cell line was not tested for contamination since correct protein was produced and validated. |
| Commonly misidentified lines<br>(See ICLAC register) | Not a commonly misidentified line. |

