## [Peer Review File · Nature Structural & Molecular Biology]

Peer Review Information

Manuscript Title: A shape-shifting nuclease unravels structured RNA

Corresponding author name(s): Leemor Joshua-Tor

Reviewer Comments & Decisions:

Decision Letter, initial version:

Message: 10th Jan 2022

Dear Professor Joshua-Tor,

Thank you again for submitting your manuscript "A shape-shifting nuclease unravels structured RNA". We now have comments (below) from the 3 reviewers who evaluated your paper. In light of those reports, we remain interested in your study and would like to see your response to the comments of the referees, in the form of a revised manuscript.

You will see that the referees raised concerns about the RNA substrates used and whether they sufficiently represent intermediate degradation states, and also requested further mutations of Dis3l2 and kinetic analysis on different hairpin substrates. Please be sure to address/respond to all concerns of the referees in full in a point-by-point response and highlight all changes in the revised manuscript text file.

We appreciate the requested revisions are extensive. We thus expect to see your revised manuscript within 6 months. If you cannot send it within this time, please let us know. We will be happy to consider your revision as long as nothing similar has been accepted for publication at NSMB or published elsewhere. Should your manuscript be substantially delayed without notifying us in advance and your article is eventually published, the received date would be that of the revised, not the original, version.

Reporting Summary:

Please note that all key data shown in the main figures as cropped gels or blots should be presented in uncropped form, with molecular weight markers. These data can be aggregated into a single supplementary figure. While these data can be displayed in a relatively informal style, they must refer back to the relevant figures. These data should be submitted with the last revision, prior to acceptance, but you may want to start putting it together at this point.

We require deposition of coordinates (and, in the case of crystal structures, structure factors) into the Protein Data Bank with the designation of immediate release upon publication (HPUB). Electron microscopy-derived density maps and coordinate data must be deposited in EMDB and released upon publication. Deposition and immediate release of NMR chemical shift assignments are highly encouraged. Deposition of deep sequencing and microarray data is mandatory, and the datasets must be released prior to or upon

publication. To avoid delays in publication, dataset accession numbers must be supplied with the final accepted manuscript and appropriate release dates must be indicated at the galley proof stage. Please find the complete NRG policies on data availability at <http://www.nature.com/authors/policies/availability.html>.

[Redacted]

Sincerely,
Sara

Sara Osman, Ph.D.
Associate Editor
Nature Structural & Molecular Biology

Referee expertise:

Referee #1: CryoEM of ribonucleases

Referee #2: RNA structural biology and biochemistry

Referee #3: Ribonuclease biology

Reviewers' Comments:

Reviewer #1:

Remarks to the Author:

RNA turnover that determines the lifetime of RNAs is a vital post-transcriptional process to control both coding and non-coding gene expression. Dis3l2 is a processive 3'-5' exonuclease that preferentially degrades 3'-uridylylated RNAs including the let-7 miRNA and other important non-coding RNAs. The Dis3l2 mediated RNA degradation holds a special

place as it can mediate an alternative mRNA decay pathway independent of the exosome. Here in this manuscript, Katarina et al presents a series of Dis3l2 structures complexed with different structured RNA, combined with the biochemical and kinetic assays, the authors proposed a novel mechanism of structured RNA degradation by dis3l2. In general, the structure combined with the kinetic data have provided some novel insights into the catalytic mechanism of nuclease Dis3l2. However there are a few issues which need to be addressed.

Major Points:

1. One of the main points of the manuscript is the dramatic conformational change induced by the substrate RNA shortening as seen in HairpinD-U7 structure. It is highly important to assess the data quality of this map so the author need better to show the masked density maps side-by-side with models of CSD1-CSD2 region in both HairpinD-U7 and RNA-free Dis3l2. The Extended Fig.4b should be modified accordingly as it only shows the RNA-free Dis3l2 could not fit in the HairpinD-U7. Similarly, It would be better to include a supplementary figure to show the representative local maps of the core regions discussed in the manuscript including the S1 and RNB loop (Extended Fig.4g,h),the RNB-trihelix-linker (Extended Fig.5c,d).
2. Does the HairpinD-U7 really reflects the true intermediated structure of the discrete degradation steps, as the authors have modified the Hairpin? Do the length of the hairpin affect the processivity or the conformational change? And following sequential degradation and the dsRNA strand separation, the length of the hairpin will decrease and moreover the junction between the dsRNA and ssRNA would not be exactly a hairpin strictly following a stretch of U as HairpinD-U7 showed, does the newly separated and released G or C bases in the junction between dsRNA and ssRNA influence the processivity or the conformational change? So The authors need to show a supplementary figure which covers all the RNA substrates of predicted RNA-fold (such as hairpin-U8, U9,U9 and U5) used in his study. Moreover the authors better show another high-resolution prong conformation structure (such as U8,U9 or U5)to illustrate the conformational change during the degradation process.
3. To capture the real snapshots of the degradation process catalyzed by dis3l2, did the authors tried to mix the wildtype active dis3l2 with the substrate HairpinA-GCU14, and prepare the time-course cryo-EM grid and collect data to capture the intermediate reaction state?

Minor Points:

1. Line 98-99, Extended Fig. 3e-g, please label "the loop" in the figures which have been mentioned in the text. And For line 99, it is better to prepare a figure to show this.
2. Comparing the Structure of Hairpin-CGU14 and U12, does the conformation of the enzyme change?
3. Line 128, Please label the RNB helices and linker in the figures.

Reviewer #2:

Remarks to the Author:

Properly regulated and efficient RNA decay is critical for cellular homeostasis, and dysregulation or dysfunction in these processes is associated with many disease states. RNA decay is the responsibility of several enzymes that must often deal with a variety of RNA substrates, to include those that are base-paired or form other stable structures.

Here, the authors explore the structural basis of RNA-degrading enzyme Dis3-like protein 2 (Dis3L2), a 3' to 5' exoribonuclease that can degrade both unpaired substrates and Watson-Crick base-paired RNA helices. Employing cryoEM, the authors solve the structure of Apo Dis3L2 and several complexes with RNA. They observed a dramatic change in the conformation of the enzyme when it engages with base-paired RNA. They then use these structures to guide and help interpret kinetic analysis of degradation on various substrates and with both mutant and wild-type enzyme. The result is a comprehensive model for the degradation of RNA (Watson-Crick paired and unpaired) by Dis3L2 that includes several structural changes.

Overall, this is a very beautifully executed study that gives very real insight and is clearly a substantial contribution to the field. The topic is important, the data are solid, the interpretations are rigorous, and the conclusions are well-supported. It is a "satisfying" example of how structure and accompanying functional studies merge to yield mechanistic models. I have very few concerns, and those that I do have are minor and listed below.

1. This is perhaps more of an editorial comment, but I thought that some of the data, lovely structural figures, and discussion in both the SI and ED could be placed into the main text and figures. I found myself flipping around an inordinate amount between 3 different things (main text, SI, and ED). These items really are important for the story and really would make the work more cohesive. I suspect that the manuscript was prepared for a different journal and thus was kept short and to 4 display items. I think the story deserves more space and figures, which should be possible in NSMB.

2. Figure 4 (the model) is nice, but I was craving a figure in which the steps of the process were shown with more of the structural details that are described in the text. These details are present, but across multiple ED figures. Again, I was flipping around a lot to see what I wanted to see. A large figure that shows the mechanism from start to finish, with structural details, would be very satisfying (think: the movie but in figure format).

3. ED Figure 1 is very low resolution.

4. Line 364: "Hepes" should be capitalized. Also, I was surprised that no Mg^{2+} was included in the RNA annealing protocol or in the cryoEM of enzyme+RNA, while Na^{+} was included. Na^{+} is not really a biologically relevant ion for RNA (K^{+} is the biologically correct monovalent cation), while Mg^{2+} is critical for RNA structure and often for its interactions with proteins. In this case, I do not think it makes a difference, but was there a reason for this choice? Related, the amount of Mg^{2+} in the kinetic analysis was 0.1 mM, which is not unreasonable, but is an order of magnitude below what is generally considered biologically relevant (1-5 mM). Again, perhaps some explanation?

5. The analysis in which the populations of particles was binned by state is interesting, and in fact there have been discussions in my lab about using similar analysis to quantitate dynamics in macromolecules. However, a concern is that to be fully rigorous, this analysis requires that there is no bias in the initial picking towards particles in particular states. In other words, can we be sure that the dataset of initial picked particles truly represents ALL the macromolecules in the micrographs, or at least is truly representative of the population distribution? Does subsequent analysis preferentially eliminate some particles in a certain state in an undetectable way (for example, when they are selected based on "high resolution"? What if one particle orientation makes the two states look identical...how are those then sorted? I did not see a reference for this type of analysis, so is this a new type of analysis (I have not seen it before). To be fair, I think

the conclusions are good, but perhaps this can be discussed a bit and any assumptions directly stated and justified.

6. I know common usage is “double-stranded,” but I prefer the term “Watson-Crick paired” to “double-stranded” for RNA duplexes of this type. This is perhaps a personal bias of mine! Except for viral replication intermediates, almost all RNA exists as a single strand, but that single strand can then interact with itself in base-paired or unpaired states. In this case, the pairing is purely Watson-Crick, but other pairings are common. Hence the most precise term is to say that the authors have provided a mechanism for how Dis3L2 can degrade stretches of “Watson-Crick paired RNA.” Likewise, the title might be a bit misleading, as the authors certainly have not shown how the enzyme might (or if it can) deal with the full gamut of RNA structure, including more complex and stable elements. I suggest a more specific and descriptive title.

Good work!

Reviewer #3:

Remarks to the Author:

This manuscript by Meze et al., entitled ‘A shape-shifting nuclease unravels structured RNA’ describes the dynamic cryo-EM structure of the RNA exonuclease DIS3L2 with different RNA substrates. DIS3L2 deficiency is responsible for the severe developmental phenotypes of Perlman syndrome and mutations in the DIS3L2 gene are linked to cancers including Wilms tumor in children. It was previously reported that DIS3L2 degrades certain RNAs that are tagged for decay by an oligo-U tail. Previous work from the Joshua-Tor lab showed the structural basis for the selective recognition and degradation of oligouridylated RNA substrates by DIS3L2. However, it remains unknown how DIS3L2 protein alone is capable of degrading structured RNA substrates. This study, utilizing structural and biochemical approaches provides some new insight into this question. In particular, the authors report a conformational change in DIS3L2 involving the repositioning of its cold shock domains (CSDs) and a wedge conformation that is specifically required for double-stranded RNA (dsRNA) decay.

The authors present 1) Cryo-EM structure of RNA-free human DIS3L2 protein at an overall 3.4 angstrom resolution. This structure is very similar to the previously reported crystal structure of mouse Dis3l2 protein bound to an oligo-U RNA.

2) 3.1 angstrom cryo-EM structure of human DIS3L2 protein bound to an oligouridylated (U14) short hairpin RNA. This structure is very similar to the previously reported mouse Dis3l2 protein/oligo-U RNA structure.

3) 3.1 angstrom cryo-EM structure of human DIS3L2 protein bound to an oligouridylated short hairpin RNA with a shorter U-tail (U12).

4) 2.8 angstrom cryo-EM structure of human DIS3L2 protein bound to an oligouridylated short hairpin RNA with a shortened U-tail (U7). This structure reveals a substantial rearrangement of the DIS3L2 protein to accommodate the dsRNA with the short oligo-U tail.

5) Subsequent cryo-EM data of Dis3L2 with a series of substrates of varying ss overhang lengths

revealed that the DIS3L2 structural rearrangement occurs as the oligo-U tail is shortened to 8-nt or shorter.

6) Biochemical experiments performed with RNA substrates with varying oligo-U tails and with wt and mutant DIS3L2 protein implicate a trihelix linker module in ds but not ss-RNA

degradation.

Overall, this study provides some new insight into the mechanism by which DIS3L2 degrades structured oligouridylated RNA substrates and in particular highlights the conformational rearrangement of DIS3L2 protein that occurs upon the transition from the ss- to ds- region of its' RNA substrate. Since mutations in DIS3L2 are linked to human disease, this study could appeal to the broader audience. As such, this work could be suitable for publication in NSMB.

Specific concerns:

- 1) Support for the model utilizes a relatively large deletion of DIS3L2 (residues P612-M669: Δ 123H). It would be more convincing to also include some more focused mutations in DIS3L2 and show how these influence degradation of ss- versus ds-RNA.
- 2) The study relies on a single hairpin RNA substrate. It will be important to include biochemical analysis of other known structured RNAs substrates of DIS3L2 to explore the generality of these findings.

Author Rebuttal to Initial comments

Response to Reviewers

We thank all of the reviewers for their detailed reading of our manuscript and for the constructive comments. These are addressed in the manuscript with updated text, figures and additional experiments. Further explanations are provided in our point-by-point responses directly below.

Reviewer #1:

Remarks to the Author:

RNA turnover that determines the lifetime of RNAs is a vital post-transcriptional process to control both coding and non-coding gene expression. Dis3l2 is a processive 3'-5' exonuclease that preferentially degrades 3'-uridylylated RNAs including the let-7 miRNA and other important non-coding RNAs. The Dis3l2 mediated RNA degradation holds a special place as it can mediate an alternative mRNA decay pathway independent of the exosome. Here in this manuscript, Katarina et al presents a series of Dis3l2 structures complexed with different structured RNA, combined with the biochemical and kinetic assays, the authors proposed a novel mechanism of structured RNA degradation by dis3l2. In general, the structure combined with the kinetic data have provided some novel insights into the catalytic mechanism of nuclease Dis3l2. However there are a few issues which need to be addressed.

Major Points:

1. One of the main points of the manuscript is the dramatic conformational change induced by the substrate RNA shortening as seen in HairpinD-U7 structure. It is highly important to assess the data quality of this map so the author need better to show the masked density maps side-by-side with models of CSD1-CSD2 region in both HairpinD-U7 and RNA-free Dis3L2. The Extended Fig.4b should be modified accordingly as it only shows the RNA-free Dis3L2 could not fit in the HairpinD-U7. Similarly, It would be better to include a supplementary figure to show the representative local maps of the core regions discussed in the manuscript including the S1 and RNB loop (Extended Fig.4g,h), the RNB-trihelixlinker (Extended Fig.5c,d).

We agree with the reviewer. We show the overlay of the density obtained from the HairpinD-U₇ structure with the RNA-free structure in the body of the manuscript in Figure 4d. We also now provide an additional figure panel in the Extended data showing the overlay of the density and the CSD1-CSD2 structure for the RNA-free and HairpinD-U₇ Dis3L2 (Extended Data Fig. 4h and i respectively). Masked density maps of the hairpinD-U₇ are shown in Extended Data Fig. 4b, and a close up of the base of the stem with density overlay is provided in Fig. 5c showing the RNB-trihelix-linker. Density of the S1 and trihelix and linker are provided in Extended data 4g, and Extended data 5c.

2. Does the HairpinD-U7 really reflects the true intermediated structure of the discrete degradation steps, as the authors have modified the Hairpin? Do the length of the hairpin affect the processivity or the conformational change? And following sequential degradation and the dsRNA strand separation, the length of the hairpin will decrease and moreover the junction between the dsRNA and ssRNA would not be exactly a hairpin strictly following a stretch of U as HairpinD-U7 showed, does the newly separated and released G or C bases in the junction between dsRNA and ssRNA influence the processivity or the conformational change? So The authors need to show a supplementary figure which covers all the RNA substrates of predicted RNA-fold (such as hairpin-U8, U9,U9 and U5) used in his study. Moreover the authors better show another high-resolution prong conformation structure (such as U8,U9 or U5)to illustrate the conformational change during the degradation process.

While it is certainly interesting to ask how different hairpin sequences or various secondary structures are degraded by Dis3L2, we chose a simple representative double-stranded RNA to interrogate steps along the degradation pathway. In order to see the elements of the structure using cryoEM at high resolution we had to shorten the hairpin. Short of trying to capture steps of the reaction in real time, the substrates used in our study enable the interrogation of these otherwise transient states at highresolution. Furthermore, we also collected data on a less GC rich stem (HairpinE-7U) and now show that the 3D reconstruction also produces the prong conformation, however at a lower resolution (Extended data Fig. 4e, f). We also see the prong conformation with two other substrates, one with a -U₈ overhang (Figure a below showing one representative class of the heterogeneous refinement in CryoSPARC), and one with a shorter tail -U₅ (Figure b below showing the homogeneous refinement in cryoSPARC). The hairpinD-U₉ complex dataset does not show the prong conformation (Figure 6).

The kinetic analyses complement the structural studies, as it shows that the substrates used for the structural studies are degraded efficiently. In order to expand our findings to the more diverse range of substrates, we added to our kinetic analysis three additional substrates with more complex structures, as well as a longer hairpin, as also requested by another reviewer.

Based on our kinetic assays, we see that after the initial slow step, Dis3L2 quickly reaches a high processivity phase which does not appear to be dependent on sequence or hairpin length. The processivity profiles for the degradation of HairpinA-GCU14 (7 bp hairpin) and a new substrate that we have tested, HairpinI-GCU14 (10 bp hairpin), are remarkably similar suggesting that hairpin length does not impact processivity or the conformational change step. We have noticed small dips 5-nucleotides away from the stable tetraloop structure; however, the change is slight and we therefore refrain from making strong conclusions about this observation.

We conclude that the timing of the conformational change depends on the RNA secondary structure (such as the complementary strand) encountering a loop in CSD1: a steric clash with the CSD1 loop triggers the repositioning of the CSDs. Therefore, it is the length of the overhang and not the hairpin that influences the timing of the large conformational change.

The conformational change takes place prior to unwinding, as shown in the 7U structure where the hairpin is still fully base-paired (Fig. 4 b, c). Our kinetic data does not show that there is a correlation between length unwound and the processivity after. There may be interactions between the freed 5' strand and the protein surface, however our data does not allow us to make conclusions about the effect of the released sequence.

We have added all the RNA fold predictions to the relevant main and extended data figures (see Fig 2a, 3a, 4a, 7a, Extended data Fig. 4a, 8c, 8d, 8e).

3. To capture the real snapshots of the degradation process catalyzed by dis3l2, did the authors tried to mix the wildtype active dis3l2 with the substrate HairpinA-GCU14, and prepare the time-course cryo-EM grid and collect data to capture the intermediate reaction state?

Due to the nature of the reaction (the high speed of degradation especially during the processive phase) a time course using available grid preparation procedures would not work. Though a couple of laboratories are developing methods for time-resolved high-resolution cryoEM, these are not yet available. However, we believe we can draw the conclusions we are making in this work based on the data we present.

Minor Points:

1. Line 98-99, Extended Fig. 3e-g, please label “the loop” in the figures which have been mentioned in the text. And For line 99, it is better to prepare a figure to show this.

We have added a label to show clearly the loop we are referring to (now Fig. 3d), and the text has been clarified.

2. Comparing the Structure of Hairpin-CGU14 and U12, does the conformation of the enzyme change? The conformation of the enzyme is the same (RMSDs, reported now on line 121, are 0.56 Å between the complexes with hairpinC-U₁₂ and hairpinA-GCU₁₄ and 0.61 Å between the complex with hairpinC-U₁₂ and the RNA-free Dis3L2). We show a comparison of the structures that shows that they have the same conformation, but we have also added a statement in the text.

3. Line 128, Please label the RNB helices and linker in the figures.

We have added a label for the trihelix liker (Fig. 5a, b).

Reviewer #2:

Remarks to the Author:

Properly regulated and efficient RNA decay is critical for cellular homeostasis, and dysregulation or dysfunction in these processes is associated with many disease states. RNA decay is the responsibility of several enzymes that must often deal with a variety of RNA substrates, to include those that are basepaired or form other stable structures. Here, the authors explore the structural basis of RNA-degrading enzyme Dis3-like protein 2 (Dis3L2), a 3' to 5' exoribonuclease that can degrade both unpaired substrates and Watson-Crick base-paired RNA helices. Employing cryoEM, the authors solve the structure of Apo Dis3L2 and several complexes with RNA. They observed a dramatic change in the conformation of the enzyme when it engages with base-paired RNA. They then use these structures to guide and help interpret kinetic analysis of degradation on various substrates and with both mutant and wild-type enzyme. The result is a comprehensive model for the degradation of RNA (Watson-Crick paired and unpaired) by Dis3L2 that includes several structural changes.

Overall, this is a very beautifully executed study that gives very real insight and is clearly a substantial contribution to the field. The topic is important, the data are solid, the interpretations are rigorous, and the conclusions are well-supported. It is a “satisfying” example of how structure and accompanying functional studies merge to yield mechanistic models. I have very few concerns, and those that I do have are minor and listed below.

1. This is perhaps more of an editorial comment, but I thought that some of the data, lovely structural figures, and discussion in both the SI and ED could be placed into the main text and figures. I found myself flipping around an inordinate amount between 3 different things (main text, SI, and ED). These items really are important for the story and really would make the work more cohesive. I suspect that the manuscript was prepared for a different journal and thus was kept short and to 4 display items. I think the story deserves more space and figures, which should be possible in NSMB.

We agree with the reviewer, and we have moved many of the Extended data figures and panels to the main text. We eliminated the supplementary discussion sections in the ED and moved them to the main text as well.

2. Figure 4 (the model) is nice, but I was craving a figure in which the steps of the process were shown with more of the structural details that are described in the text. These details are present, but across multiple ED figures. Again, I was flipping around a lot to see what I wanted to see. A large figure that shows the mechanism from start to finish, with structural details, would be very satisfying (think: the movie but in figure format).

We have changed to a less cartoon-like representation, using instead a space-filling model of the structures. We did however feel that this final summary benefited from a simplified representation to preserve clarity, especially for a reader with less structural expertise.

3.ED Figure 1 is very low resolution.

We thank the reviewer for drawing our attention to the formatting problem, which we have now fixed.

4. Line 364: “Hepes” should be capitalized. Also, I was surprised that no Mg^{2+} was included in the RNA annealing protocol or in the cryoEM of enzyme+RNA, while Na^+ was included. Na^+ is not really a biologically relevant ion for RNA (K^+ is the biologically correct monovalent cation), while Mg^{2+} is critical for RNA structure and often for its interactions with proteins. In this case, I do not think it makes a difference, but was there a reason for this choice? Related, the amount of Mg^{2+} in the kinetic analysis was 0.1 mM, which is not unreasonable, but is an order of magnitude below what is generally considered biologically relevant (1-5 mM). Again, perhaps some explanation?

The presence of Mg^{2+} is not crucial for folding of short hairpins and it can encourage base hydrolysis at high temperatures, therefore we omitted it during annealing. The reviewer is of course correct regarding the low Mg^{2+} concentrations for the kinetic analysis, however higher concentrations of Mg^{2+} would speed up the reaction making it more difficult to obtain accurate kinetic parameters, and therefore less practical. We corrected the capitalization for HEPES, we thank the reviewer for pointing that out.

5. The analysis in which the populations of particles was binned by state is interesting, and in fact there have been discussions in my lab about using similar analysis to quantitate dynamics in macromolecules. However, a concern is that to be fully rigorous, this analysis requires that there is no bias in the initial picking towards particles in particular states. In other words, can we be sure that the dataset of initial picked particles truly represents ALL the macromolecules in the micrographs, or at least is truly representative of the population distribution? Does subsequent analysis preferentially eliminate some particles in a certain state in an undetectable way (for example, when they are selected based on “high resolution”? What if one particle orientation makes the two states look identical...how are those then sorted? I did not see a reference for this type of analysis, so is this a new type of analysis (I have not seen it before). To be fair, I think the conclusions are good, but perhaps this can be discussed a bit and any assumptions directly stated and justified.

We have also spent some time thinking about the classification of states and the processing protocol. All datasets were picked using the same Warp neural-network based picker, BoxNet2Mask_20180918. This picker should distinguish between particles, background, and high contrast artifacts. Bad picks are removed if they are located on or close to the artifacts, if they are too close to another particle. While a small number of particles may be missed by any picking program, we believe that for the purpose of our comparison, which is simply evaluating the presence or absence of the prong or vase, the Warp picker is reliable. We are not drawing conclusions based on levels of either in any substantial way, but a very qualitative, high-level comparison.

To increase our confidence in this qualitative analysis, and to check if the selection at the stage of 2D classification was introducing significant bias, we re-processed a dataset (hairpinC-U₁₂) by using all of the 'good-particles' from Warp (in other words, we skipped the 2D classification and selection of good classes step) and obtained a similar proportion of particles representing a "vase" reconstruction (~70%) to that seen in the conventional protocol (~80%) (Extended data Fig. 6d, e).

6. I know common usage is "double-stranded," but I prefer the term "Watson-Crick paired" to "doublestranded" for RNA duplexes of this type. This is perhaps a personal bias of mine! Except for viral replication intermediates, almost all RNA exists as a single strand, but that single strand can then interact with itself in base-paired or unpaired states. In this case, the pairing is purely Watson-Crick, but other pairings are common. Hence the most precise term is to say that the authors have provided a mechanism for how Dis3L2 can degrade stretches of "Watson-Crick paired RNA." Likewise, the title might be a bit misleading, as the authors certainly have not shown how the enzyme might (or if it can) deal with the full gamut of RNA structure, including more complex and stable elements. I suggest a more specific and descriptive title.

We appreciate the reviewer's opinion. First, the types of substrates that this enzyme largely encounters are a slew of uridylylated non-coding RNAs, such as RMRP, 7SL, many of which have quite complex structures and interactions, and many of which include many other pairings rather than strictly WatsonCrick only pairings. Second, even in the example we present here, not all of the RNA is comprised of Watson-Crick pairs (e.g., the base of the stem in hairpinA-GCU₁₄ is a wobble-pair: GU). Therefore, we decided to stick to double-stranded for simplicity. We did not deal with the whole gamut of structures, but now included additional ones, though our title does not imply a universal mechanism for all structured RNAs.

Good work!

Thank you!

Reviewer #3:

This manuscript by Meze et al., entitled 'A shape-shifting nuclease unravels structured RNA' describes the dynamic cryo-EM structure of the RNA exonuclease DIS3L2 with different RNA substrates. DIS3L2 deficiency is responsible for the severe developmental phenotypes of Perlman syndrome and mutations in the DIS3L2 gene are linked to cancers including Wilms tumor in children. It was previously reported that DIS3L2 degrades certain RNAs that are tagged for decay by an oligo-U tail. Previous work from the Joshua-Tor lab showed the structural basis for the selective recognition and degradation of oligouridylylated RNA substrates by DIS3L2. However, it remains unknown how DIS3L2 protein alone is capable of degrading structured RNA substrates. This study, utilizing structural and biochemical

approaches provides some new insight into this question. In particular, the authors report a conformational change in DIS3L2 involving the repositioning of its cold shock domains (CSDs) and a wedge conformation that is specifically required for double-stranded RNA (dsRNA) decay. The authors present 1) Cryo-EM structure of RNA-free human DIS3L2 protein at an overall 3.4 angstrom resolution. This structure is very similar to the previously reported crystal structure of mouse Dis3l2 protein bound to an oligo-U RNA.

2) 3.1 angstrom cryo-EM structure of human DIS3L2 protein bound to an oligouridylated (U14) short hairpin RNA. This structure is very similar to the previously reported mouse Dis3l2 protein/oligo-U RNA structure.

3) 3.1 angstrom cryo-EM structure of human DIS3L2 protein bound to an oligouridylated short hairpin RNA with a shorter U-tail (U12).

4) 2.8 angstrom cryo-EM structure of human DIS3L2 protein bound to an oligouridylated short hairpin RNA with a shortened U-tail (U7). This structure reveals a substantial rearrangement of the DIS3L2 protein to accommodate the dsRNA with the short oligo-U tail.

5) Subsequent cryo-EM data of Dis3L2 with a series of substrates of varying ss overhang lengths revealed that the DIS3L2 structural rearrangement occurs as the oligo-U tail is shortened to 8-nt or shorter.

6) Biochemical experiments performed with RNA substrates with varying oligo-U tails and with wt and mutant DIS3L2 protein implicate a trihelix linker module in ds but not ss-RNA degradation.

Overall, this study provides some new insight into the mechanism by which DIS3L2 degrades structured oligouridylated RNA substrates and in particular highlights the conformational rearrangement of DIS3L2 protein that occurs upon the transition from the ss- to ds- region of its' RNA substrate. Since mutations in DIS3L2 are linked to human disease, this study could appeal to the broader audience. As such, this work could be suitable for publication in NSMB.

Specific concerns:

1) Support for the model utilizes a relatively large deletion of DIS3L2 (residues P612-M669: Δ 123H). It would be more convincing to also include some more focused mutations in DIS3L2 and show how this influence degradation of ss- versus ds-RNA.

We have tested individual deletions of individual helices that make up the RNB trihelix. We found that deletion of any individual helix from this bundle did not abolish the enzyme's ability to degrade ds-RNA (now added as Extended data Fig. 10k). We believe that the reason for this is that it is the whole structural module that forms the wedge rather than particular side chains or even one of the helices.

2) The study relies on a single hairpin RNA substrate. It will be important to include biochemical analysis of other known structured RNAs substrates of DIS3L2 to explore the generality of these findings.

We appreciate this comment and therefore we designed three additional ds-RNA substrates, two of which were mimics of the natural substrate 7SL (hairpinF-U₁₆, hairpinG-U₁₆), and one with a longer stem (hairpinI-GCU₁₄) compared to hairpinA-GCU₁₄. Representative gels of the titration reactions and the pulse-chase experiments for each RNA are shown in panels Extended data Fig. 8a and b. The fold of these is shown in Extended data Fig. 8c, d, e along with their respective calculated processivity values. The results show that while there are small differences in the processivity of the first step, the overall profile is very similar to hairpinA-GCU₁₄. The results are described in more detail in the main text.

Decision Letter, first revision:

Message: Our ref: NSMB-A45626A

10th Aug 2022

Dear Dr. Joshua-Tor,

Thank you for submitting your revised manuscript "A shape-shifting nuclease unravels structured RNA" (NSMB-A45626A). It has now been seen by the original referees and their comments are below. The reviewers find that the paper has improved in revision, and therefore we'll be happy in principle to publish it in Nature Structural & Molecular Biology, pending minor revisions to satisfy the referees' final requests and to comply with our editorial and formatting guidelines.

We are now performing detailed checks on your paper and will send you a checklist detailing our editorial and formatting requirements in about 3 weeks. Please do not upload the final materials and make any revisions until you receive this additional information from us.

To facilitate our work at this stage, we would appreciate if you could send us the main text as a word file. Please make sure to copy the NSMB account (cc'ed above).

Sincerely,
Sara

Sara Osman, Ph.D.
Associate Editor
Nature Structural & Molecular Biology

Reviewer #1 (Remarks to the Author):

The authors have addressed all the questions I raised and the manuscript has been revised better now. I have no further comments and it is now suitable for publication.

Reviewer #2 (Remarks to the Author):

Thank you for the opportunity to re-review this important work. My initial review was positive, with only a few minor comments meant to improve the work. The authors have addressed all of those, and it should be published in NSMB ASAP.

Reviewer #3 (Remarks to the Author):

In this revised manuscript the authors have adequately addressed previous comments and the new data further strengthen their conclusions. This revision is therefore considered acceptable for publication in NSMB.

Decision Letter, author guidance

Message: Our ref: NSMB-A45626A

11th Oct 2022

Dear Dr. Joshua-Tor,

Thank you for your patience as we've prepared the guidelines for final submission of your Nature Structural & Molecular Biology manuscript, "A shape-shifting nuclease unravels structured RNA" (NSMB-A45626A). Please carefully follow the step-by-step instructions provided in the attached file, and add a response in each row of the table to indicate the changes that you have made. Please also check and comment on any additional marked-up edits we have proposed within the text. Ensuring that each point is addressed will help to ensure that your revised manuscript can be swiftly handed over to our production team.

We would like to start working on your revised paper, with all of the requested files and forms, as soon as possible. If you can resubmit within the next week it is possible that your submission could be published before the end of 2022. Please get in contact with us if you anticipate any delays in resubmission.

In recognition of the time and expertise our reviewers provide to Nature Structural & Molecular Biology's editorial process, we would like to formally acknowledge their contribution to the external peer review of your manuscript entitled "A shape-shifting

nuclease unravels structured RNA". For those reviewers who give their assent, we will be publishing their names alongside the published article.

Nature Structural & Molecular Biology offers a Transparent Peer Review option for new original research manuscripts submitted after December 1st, 2019. As part of this initiative, we encourage our authors to support increased transparency into the peer review process by agreeing to have the reviewer comments, author rebuttal letters, and editorial decision letters published as a Supplementary item. When you submit your final files please clearly state in your cover letter whether or not you would like to participate in this initiative. Please note that failure to state your preference will result in delays in accepting your manuscript for publication.

Cover suggestions

As you prepare your final files we encourage you to consider whether you have any images or illustrations that may be appropriate for use on the cover of Nature Structural & Molecular Biology.

Nature Structural & Molecular Biology has now transitioned to a unified Rights Collection system which will allow our Author Services team to quickly and easily collect the rights and permissions required to publish your work. Approximately 10 days after your paper is formally accepted, you will receive an email in providing you with a link to complete the grant of rights. If your paper is eligible for Open Access, our Author Services team will also be in touch regarding any additional information that may be required to arrange payment for your article.

Please note that *Nature Structural & Molecular Biology* is a Transformative Journal (TJ). Authors may publish their research with us through the traditional subscription access route or make their paper immediately open access through payment of an article-processing charge (APC). Authors will not be required to make a final decision about access to their article until it has been accepted. [Find out more about Transformative Journals](https://www.springernature.com/gp/open-research/transformative-journals)

Authors may need to take specific actions to achieve  > **compliance** **with funder and institutional open access mandates.** If your research is supported by a funder that requires immediate open access (e.g. according to [Plan S principles](https://www.springernature.com/gp/open-research/plan-s-compliance)) then you should select the gold OA route, and we will direct you to the compliant route where possible. For authors selecting the subscription publication route, the journal's standard licensing terms will need to be accepted, including [self-archiving policies](https://www.nature.com/nature-portfolio/editorial-policies/self-archiving-and-license-to-publish). Those licensing terms will supersede any other terms that the author or any third party may assert apply to any version of the manuscript.

Please use the following link for uploading these materials:
[Redacted]

Best regards,

Sophia Frank
Editorial Assistant
Nature Structural & Molecular Biology
nsmb@us.nature.com

On behalf of

Sara Osman, Ph.D.
Associate Editor
Nature Structural & Molecular Biology

Reviewer #1:

Remarks to the Author:

The authors have addressed all the questions I raised and the manuscript has been revised better now. I have no further comments and it is now suitable for publication.

Reviewer #2:

Remarks to the Author:

Thank you for the opportunity to re-review this important work. My initial review was positive, with only a few minor comments meant to improve the work. The authors have addressed all of those, and it should be published in NSMB ASAP.

Reviewer #3:

Remarks to the Author:

In this revised manuscript the authors have adequately addressed previous comments and the new data further strengthen their conclusions. This revision is therefore considered acceptable for publication in NSMB.

Final Decision Letter:

Message 11th Jan 2023

:

Dear Dr. Joshua-Tor,

We are now happy to accept your revised paper "A shape-shifting nuclease unravels structured RNA" for publication as an Article in Nature Structural & Molecular Biology.

To assist our authors in disseminating their research to the broader community, our SharedIt initiative provides all co-authors with the ability to generate a unique shareable

link that will allow anyone (with or without a subscription) to read the published article. Recipients of the link with a subscription will also be able to download and print the PDF.

Your paper will be published online soon after we receive proof corrections and will appear in print in the next available issue. You can find out your date of online publication by contacting the production team shortly after sending your proof corrections. Content is published online weekly on Mondays and Thursdays, and the embargo is set at 16:00 London time (GMT)/11:00 am US Eastern time (EST) on the day of publication. Now is the time to inform your Public Relations or Press Office about your paper, as they might be interested in promoting its publication. This will allow them time to prepare an accurate and satisfactory press release. Include your manuscript tracking number (NSMB-A45626B) and our journal name, which they will need when they contact our press office.

About one week before your paper is published online, we shall be distributing a press release to news organizations worldwide, which may very well include details of your work. We are happy for your institution or funding agency to prepare its own press release, but it must mention the embargo date and Nature Structural & Molecular Biology. If you or your Press Office have any enquiries in the meantime, please contact press@nature.com.

Please note that *Nature Structural & Molecular Biology* is a Transformative Journal (TJ). Authors may publish their research with us through the traditional subscription access

route or make their paper immediately open access through payment of an article-processing charge (APC). Authors will not be required to make a final decision about access to their article until it has been accepted. [Find out more about Transformative Journals](https://www.springernature.com/gp/open-research/transformative-journals)

Sincerely,
Sara

Sara Osman, Ph.D.
Associate Editor
Nature Structural & Molecular Biology
